# Invasive Pulmonary Aspergillosis in Patients with and without SARS-CoV-2 Infection

**DOI:** 10.3390/jof9020130

**Published:** 2023-01-17

**Authors:** Jesús Fortún, María Mateos, Elia Gómez-García de la Pedrosa, Cruz Soriano, David Pestaña, José Palacios, Javier López, Santiago Moreno

**Affiliations:** 1Infectious Diseases Department, Hospital Ramón y Cajal, 28034 Madrid, Spain; 2IRYCIS: Instituto Ramón y Cajal de Investigación Sanitaria, 28034 Madrid, Spain; 3CIBER de Enfermedades Infecciosas (CIBERINFEC), 28029 Madrid, Spain; 4Facultad de Medicina, Universidad de Alcalá, 28805 Alcalá de Henares, Spain; 5Internal Medicine Department, Hospital Infanta Cristina, 28981 Madrid, Spain; 6Microbiology Department, Hospital Ramón y Cajal, 28034 Madrid, Spain; 7Intensive Care Unit, Hospital Ramón y Cajal, 28034 Madrid, Spain; 8Anaesthesia and Reanimation Department, Hospital Ramón y Cajal, 28034 Madrid, Spain; 9Pathology Department, Hospital Ramon y Cajal, 28034 Madrid, Spain; 10CIBER de Cáncer (CIBERONC), 28029 Madrid, Spain; 11Hematology Department, Hospital Ramón y Cajal, 28034 Madrid, Spain

**Keywords:** aspergillosis, CAPA, SARS-CoV-2, colonization, thrombocytopenia

## Abstract

The recent European Confederation of Medical Mycology (ECMM) and the International Society for Human and Animal Mycology (ISHAM) 2020 consensus classification proposes criteria to define coronavirus 2019 (COVID-19)-associated invasive pulmonary aspergillosis (CAPA), including mycological evidence obtained via non-bronchoscopic lavage. Given the low specificity of radiological findings in patients with severe acute respiratory syndrome coronavirus 2 (SARS-CoV-2) infection, this criterion makes it difficult to differentiate between invasive pulmonary aspergillosis (IPA) and colonization. This unicenter and retrospective study includes 240 patients with isolates of any *Aspergillus* species in any respiratory samples during a 20-month study (140 IPA and 100 colonization). Mortality was high in the IPA and colonization groups (37.1% and 34.0%, respectively; *p* = 0.61), especially in patients with SARS-CoV-2 infection, where mortality was higher in colonized patients (40.7% vs. 66.6.%; *p*: 0.021). Multivariate analysis confirmed the following variables to be independently associated with increased mortality: age > 65 years, acute or chronic renal failure at diagnosis, thrombocytopenia (<100,000 platelets/µL) at admission, inotrope requirement, and SARS-CoV-2 infection, but not the presence of IPA. This series shows that the isolation of *Aspergillus* spp. in respiratory samples, whether associated with disease criteria or not, is associated with high mortality, especially in patients with SARS-CoV-2 infection, and suggests an early initiation of treatment given its high mortality rate.

## 1. Introduction

COVID-19-associated invasive pulmonary aspergillosis (CAPA) rates in patients admitted to intensive care units (ICUs) have been reported to range between 19.6% and 33.3% [1,2,3,4]. CAPA had worse outcomes as measured by ordinal severity of disease scores, requiring a longer time to improve. Mortality rates were approximately 50% [5,6,7,8,9,10,11].

There are similarities between influenza-associated pulmonary aspergillosis (IAPA) and CAPA, including high prevalence, absence of classic host factors for IPA, similar timing in the disease diagnosis after ICU admission, and the presence of lymphopenia or thrombocytopenia, but it is unclear whether severe acute respiratory syndrome itself is the main risk factor for CAPA, or whether additional risk factors, such as corticosteroid therapy or anti-interleukin (IL)-6 treatment with tocilizumab further increase the risk for disease progression [8,9,10,11,12].

Numerous studies have been published on CAPA since the COVID-19 pandemic began, and new classifications have even been proposed to better reflect the intrinsic characteristics of IPA in patients with SARS-CoV-2 infection [1]. However, few series have analyzed the clinical, diagnostic, and prognostic differences between patients with and without SARS-CoV-2 infection. The value of aspergillus colonization in severe patients with and without SARS-CoV-2 infection and the impact of including the possible CAPA category in the latest diagnostic recommendations have also not been evaluated in depth.

A diagnostic, clinical, and prognostic characteristic of patients with *Aspergillus* spp. isolates in respiratory samples in patients with and without SARS-CoV-2 infection in our center is presented. The role of *Aspergillus* colonization in these patients, the importance of the new category of possible IPA according to the ECMM/ISHAM 2020 consensus criteria [1] and in the other classifications already available [13,14,15,16], the risk factors of IPA, and the mortality rates in the different groups are analyzed.

## 2. Methods

### 2.1. Study Setting

Retrospective cohort study analyzing hospitalized adult patients, with and without SARS-CoV-2 infection, with isolation of *Aspergillus* spp. in respiratory samples, between 3 March 2020 and 30 June 2021, in a tertiary hospital in Madrid, Spain.

### 2.2. Study Design and Participants

Patients >18 years old were selected in a centralized electronic format, with anonymous data extracted by the investigator. Data were encrypted. A single online database was used. The application included epidemiological, clinical, diagnostic, imaging, therapeutic, and prognostic variables. The study was approved by the Institutional Review Board of Ramon y Cajal Hospital.

### 2.3. Inclusion/Exclusion Criteria

Hospitalized patients with isolation of any *Aspergillus* species in any of the following respiratory samples: sputum, tracheo-bronchial aspirate (TBA), or bronchoalveolar lavage (BAL), were included in the study. No mini-BAL or blind BAL was performed. In addition, a group of 31 patients with SARS-CoV-2 infection with and without IPA with an available autopsy study performed at the center during the same study period was analyzed. Patients <18 years old, or patients with an absence of *Aspergillus* spp. isolation in a respiratory specimen (except patients with IPA confirmed in the autopsy study group) were excluded.

### 2.4. Microbiological Methods

Detection of SARS-CoV-2 virus was performed by real-time RT-PCR, following the WHO and/or CDC protocol (QuantStudio S5 Real-time PCR system; ThermoFisher, Waltham, MA, USA). The galactomannan (GM) antigen index was measured with a sandwich enzyme-linked immunosorbent assay (ELISA) (Platelia™ Aspergillus ELISA, Bio-Rad, Marnes-la-Coquette, France) in BAL and serum specimens. Calcofluor white stain in BAL was performed looking for the presence of hyphae suggestive of infection by *Aspergillus* or other molds. BAL was further analyzed by culture for filamentous fungi. A 10-µL volume of BAL was cultured on Sabouraud Chloramphenicol agar tubes. As soon as molds were visible, they were subcultured on Potato Dextrose Agar plates for 2 to 3 days at 30 °C. Fungi identification was performed by microscopic examination of lactophenol cotton-blue stained slides and by MALDI-TOF Mass Spectrometry Instrument (Bruker Biotyper; Bruker Daltonics, Bremen, Germany), following the manufacturer’s instructions. Quantitative real time-PCR for *Aspergillus* was not performed.

### 2.5. Diagnostic Criteria

#### 2.5.1. Patients without SARS-CoV-2 Infection

For the diagnosis of IPA in patients without SARS-CoV-2 infection, the recently modified EORTC/MSG criteria (in immunocompromised patients) [13], the AspICU definitions for probable/putative IPA, in ICU patients [14], and the Bulpa criteria, in patients with chronic obstructive pulmonary disorder (COPD) [15] were considered. Proven IPA in these patients was like proven CAPA.

For the diagnosis of probable IPA, according to the EORTC/MSG criteria [13], the presence of at least one of the following four patterns of CT was required: dense, well-circumscribed lesions(s) with or without a halo sign, air crescent sign cavity, wedge-shaped and segmental or having a lobar consolidation, in combination with *Aspergillus* spp. isolation in respiratory samples, or a positive GM in serum or BAL, or a positive direct test (cytology, direct microscopy).

For the AspICU criteria [14], the following combination was required for the diagnosis of putative/probable IPA: (1) microbiological criteria: isolation of *Aspergillus* spp. in the lower respiratory tract, or serum positive GM (>0.5, repeated), or positive GM in BAL (>1.0); (2) compatible signs and symptoms (one of the following): (a) fever refractory to at least 3 days of appropriate antibiotic therapy, (b) recrudescent fever after a period of defervescence of at least 48 h while still on antibiotics, without other apparent causes, (c) pleuritic chest pain or rub, (d) dyspnea, (e) hemoptysis, or (f) worsening respiratory insufficiency despite appropriate antibiotic therapy and respiratory support; and (3) abnormal medical imaging by portable chest X-ray or CT scan of the lungs.

For the diagnosis of IPA in COPD patients, the following criteria were used [15]:

*Probable IPA*: GOLD (stage III or IV) with recent exacerbation of dyspnea, suggestive chest imaging, and one of the following: (1) positive culture and/or microscopy for *Aspergillus* from the lower respiratory tract (LRT); (2) positive serum antibody test for *A. fumigatus* (including precipitins); (3) two consecutive positive serum GM tests.

*Possible IPA*: like probable IPA but without positive *Aspergillus* culture or microscopy from LRT or serology.

#### 2.5.2. Patients with SARS-CoV-2 Infection

For the diagnosis of pulmonary aspergillosis associated with SARS-CoV-2 infection (CAPA), the modified ECMM/ISHAM 2020 consensus (*) criteria were applied [1]:

*Proven CAPA*: IPA confirmed by the histological specimen (lung biopsy or autopsy) or direct lung or tracheobronchial microscopy.

*Probable CAPA*: Compatible clinical and imaging findings (pulmonary infiltrate or nodules preferably documented by chest CT or cavitating infiltrate), with microbiological isolation of *Aspergillus* in BAL, or galactomannan (GM) in BAL >1, or GM in serum >0.5 or positive PCR in serum (not available in this study), or positive PCR in BAL (<36 cycles) (not available in this study).

*Possible CAPA*: Compatible clinical and imaging findings, with microbiological isolation of *Aspergillus* in respiratory specimens other than BAL: TBA, sputum. Possible pulmonary CAPA requires pulmonary infiltrates, well-circumscribed lesions(s) or nodules, preferably documented by chest CT, or cavitating infiltrate, which is not attributed to another cause. In patients with bilateral, ground-glass opacities or other COVID-19 related findings, significant radiological changes as previously mentioned and confirmed by an expert radiologist were required to be considered possible CAPA. The ECMM/ISHAM 2020 consensus includes non-bronchoscopic lavage as a diagnostic tool (*). Non-bronchoscopic lavage was not performed in this study In this series, these were replaced (only in patients with SARS-CoV-2 infection) with non-bronchoscopic samples: TBA, sputum.

*Probable Traqueobronquitis:* The presence of tracheobronchial ulceration, nodule, pseudomembrane, plaque, or eschar seen on a bronchoscopic analysis together with visualization of hyphae or isolation of *Aspergillus* spp. in culture.

Colonization was considered in patients with no radiological findings or unchanged with respect to those attributed to COVID-19.

In case of belonging to more than one group, they were prioritized as follows: In patients with SARS-CoV-2 infection, the modified ECMM/ISHAM 2020 consensus was always applied. In patients without SARS-CoV-2 infection, the EORTC/MSG criteria were taken into account first, then the AspICU criteria and finally the Bulpa criteria.

A separate analysis was performed according to IPA. PPP-IPA integrated the diagnoses of proven, probable/putative, and possible IPA, regardless of the criteria used; in this case, the remaining patients included in the series were considered colonized. PP-IPA integrated only the diagnoses of proven and probable IPA; in this case, the diagnoses of possible IPA were analyzed together with the colonized patients.

### 2.6. Statistical Analysis

Standard descriptive statistics were used to summarize study population characteristics. Student’s t-tests or Mann-Whitney U-tests were used for continuous variables. The association between categorical variables was evaluated using the chi-square test or Fisher’s exact test. The McNemar χ^2^ test was used to compare sensitivity, specificity, negative predictive value (NPV), and positive predictive value (PPV) among diagnostic procedures. Logistic regression was used to identify mortality predictors. The Hosmer-Lemeshow test assessed the goodness of fit for logistic regression models. All statistical studies were performed using IBM^®^ SPSS^®^ Statistics version 25 (Armonk, NY, USA: IBM Corp).

## 3. Results

### 3.1. Characteristics of Patients with Isolation of Aspergillus *spp.*

During the 20-month study period, 240 patients with isolates of any *Aspergillus* species in any of the respiratory samples [sputum, tracheo-bronchial aspirate (TBA), or bronchoalveolar lavage (BAL)] were analyzed. According to the type of patients and different criteria, 100 were considered colonizations and 140 were considered proven, probable/putative, or possible aspergillosis (IPA).

Table 1 shows the main characteristics of both groups. In the series, there was 3.6% of tracheobronchitis. Except for the presence of active hematologic disease and corticosteroid therapy, there were no significant differences in the baseline conditions of the two groups. There were also no differences in the main analytical data at admission, but respiratory requirements and the need for ICU, inotropes, and days of ICU or hospitalization were higher in patients with IPA. Although there was a higher frequency of SARS-CoV-2 infection in patients with IPA (77.1%), it was present in 24% of colonized patients.

There were more colonized patients in SARS-CoV-2 group (76 colonized and 32 IPA) than in patients without SARS-CoV-2 infection (24 colonized, 108 IPA). Colonized patients did not show lower mortality, and mortality was comparable with IPA (37.1% and 34.0%, respectively). Notably, in patients with SARS-CoV-2 infection, mortality was higher in colonized patients (16/24, 66.6%) than in IPA (44/108 40.7%) (*p*: 0.021). However, in patients without SARS-CoV-2 infection, mortality in colonized patients was 13.2% (10/76) and 25% (8/32) in the IPA group (*p* = 0.16). When possible IPA was excluded and only proved/probable or putative IPA was considered (PP-IPA), among colonized patients the mortality rate remained high (50%) in patients with SARS-CoV-2 infection and was lower in patients without SARS-CoV-2 infection (14.0%), similar to mentioned 13.2%.

Overall mortality was 35.8%. Table 2 shows uni- and multivariate analysis among all the variables included in Table 1. It is noteworthy that neither PPP-IPA (proven/probable/possible IPA) nor PP-IPA (only proven/probable IPA) were associated with higher mortality, confirming the poor prognosis of colonized patients; however, it was significantly higher in the SARS-CoV-2 infection group (51.5%, *p* < 0.001). The following variables resulted independently associated with a higher mortality in multivariate analysis: >65 years old (OR 2.69; CI95%: 1.29–5.57), acute or chronic renal failure at diagnosis (OR: 2.58; CI95%: 1.30–5.08), thrombocytopenia (<100,000 platelets/µL) at admission (OR: 6.41; CI95%: 2.07–19.8), inotrope requirement (OR: 5.32; CI95%: 2.52–11.19), and SARS-CoV-2 infection (OR: 3.14; 1.40–7.04).

### 3.2. Platelet Count and Mortality

Although the mean baseline platelet count was similar between IPA and colonized patients, patients with IPA more frequently had platelet counts <100,000/µL (18/138, 13.0% vs. 5/86, 5.8% in colonized patients). Mortality was higher in patients with IPA and platelet counts <100,000/µL (14/18, 77.8%) than patients with IPA and >100,000/µL (38/120, 31.7%), *p* < 0.001. Mortality was also high (3/5, 60%) in colonized patients with baseline platelet counts <100,000/µL, but there was no difference with colonized patients with baseline platelet counts >100,000/µL (30/81, 37.0%, *p* = 0.36). Low platelet counts and mortality were confirmed mainly in patients with SARS-CoV-2 infection. In these patients, mortality in patients with IPA and platelet counts <100,000/µL was elevated (10/11, 90.9%), compared to mortality in the same group with IPA and baseline platelet counts >100,000 (34/97, 35.1%, *p* < 0.001). Mortality, although high, was lower in patients without SARS-CoV-2 infection with IPA and low platelet counts (4/7, 57.1%) vs. high platelet count (4/23,17.4%, *p*: 0.06).

When multivariate analysis of mortality was restricted to patients with CAPA, baseline platelet count <100,000/µL remained an independent factor for mortality (OR: 19.0 (95%CI: 2.12–170.8), along with renal failure on admission (OR: 4.2, 95%CI: 1.67–10.5) and inotrope requirement (OR: 3.62, 95%CI: 1.35–9.70).

### 3.3. Patients with IPA

Table 3 summarizes the differences between patients with or without SARS-CoV-2 infection among IPA patients (excluding colonized). IPA in SARS-CoV-2 infection was more frequent in males; the most predominant type of infiltrate was bilateral, ground-glass opacities, required more respiratory assistance and ICU admission, and was associated with a longer duration of previous corticosteroid therapy. On the other hand, IPA in patients without SARS-CoV-2 infection had a higher number of PP-IPA; possible IPA in patients without SARS-CoV-2 infection accounted for 31.8% and was mostly in COPD patients, according to Bulpa criteria, while in patients with SARS-CoV-2 infection, possible IPA was the most common form (81.8%).

Previous aspergillosis infection was more frequent among patients without SARS-CoV-2 infection. Regarding treatment, there were no clear differences, except that isavuconazole was used more in patients with SARS-CoV-2 infection. There were no significant differences in overall mortality, although it was more frequent in patients with SARS-CoV-2 infection (40.7% vs. 25.0%).

Table 4 shows the variables associated with higher mortality among patients with IPA. As mentioned, SARS-CoV-2 infection was not associated with higher mortality in patients with IPA. Multivariate analysis showed similar results to the overall population, excluding age and SARS-CoV-2 infection, but it was confirmed for acute or chronic renal failure (OR 4.93; 95%CI: 2.12–11.5), platelet count <100,000 (OR: 7.91, 95%CI: 2.16–29.0), and inotrope requirement (OR: 4.89; 95%CI: 2.07–11.5).

Comparison between patients with or without SARS-CoV-2 infection (n: 24 vs. n: 22) in an analysis restricted only to the population PP-IPA also showed similar results. Multivariate analysis was also concordant in this group and showed the same factors independently associated with higher mortality: acute or chronic renal failure (OR: 4.53; 95%CI: 2.12–11.5), platelet count <100,000/µL (OR: 7.91; 95%CI: 2.15–29.0), and inotrope requirement (OR: 4.80; 95%CI: 2.07–11.5).

### 3.4. Patients with Autopsy Study

Table 5 shows the characteristics of 31 patients with SARS-CoV-2 infection with available necropsy studies. Of the 31 patients, only 4 showed confirmatory findings of IPA (proven). In the 27 patients in whom autopsy did not show findings suggestive of IPA, 3 had shown before death a diagnosis of IPA (2 of them PP-IPA) and had received prolonged antifungal treatment. In the four patients with confirmed IPA at autopsy, only one had shown isolation of *Aspergillus* spp. in respiratory samples (sputum) and had GM+ in BAL; however, the remaining three had shown negative results in sputum, TBA, or BAL. None of the four patients had shown GM+ in serum.

### 3.5. Efficacy of Diagnostic Techniques

Table 6 shows the sensitivity, specificity, and predictive values of the six diagnostic techniques applied in this study: sputum culture, tracheo-bronchial aspirate culture, bronchoalveolar lavage culture, presence of hyphae in BAL, GM in BAL, and GM in serum. The first two rows show results in patients with SARS-CoV-2 infection under the diagnosis of PPP-IPA, considering the modified ECMM/ISHAM 2020 consensus used in this study (first row) or PP-IPA (2nd row); rows three and four show the same but in patients without SARS-CoV-2 infection. The diagnosis in this study of possible IPA is closely related to the participation of cultures of upper respiratory samples (sputum and TBA). On this basis, sputum and TBA showed good sensitivities and positive predictive values, especially in patients with SARS-CoV-2 infection, since in these patients, the diagnosis of possible IPA was more numerous than in patients without SARS-CoV-2 infection.

The best performing technique was GM in BAL, especially for diagnosing PP-IPA. GM BAL presented PPV >88% in all groups and, except for possible IPA in patients with SARS-CoV-2 infection, achieved good negative predictive values. The low sensitivity of GM in serum is remarkable, even in patients without SARS-CoV-2 infection, which includes hematological patients with angioinvasive forms. Finally, although the presence of hyphae in BAL is a very specific technique for hyalohyphomycosis, its poor sensitivity in patients with SARS-CoV-2 infection is remarkable.

## 4. Discussion

Following the SARS-CoV-2 pandemic, several published series have highlighted the importance of *Aspergillus* superinfection in these patients [1,2,3,4,5,6,7,8,9,10,11]. However, few series have analyzed the epidemiological, diagnostic, and prognostic differences of IPA in patients with and without SARS-CoV-2 infection. 

First, this series shows that in hospitalized patients and mainly in ICU admits, the isolation of *Aspergillus* spp. in respiratory samples, with or without IPA criteria (colonization), is associated with high morbidity and mortality. In fact, the mortality analysis, including patients colonized or infected by *Aspergillus* spp., does not confirm IPA (PPP-IPA or PP-IPA) as an independent factor.

Although there are many series confirming that colonization by *Aspergillus* spp. is associated with a worse prognosis in COPD patients [17,18], cystic fibrosis [19,20], solid organ transplantation [20,21], and critically ill patients [22,23,24,25,26], no series equates the prognosis between colonized and infected patients, which is significantly worse in infected patients. The high mortality in colonized patients in this series are mainly due to patients with SARS-CoV-2 infection. The 66% mortality rate in colonized and SARS-CoV2-infected patients is striking, in contrast to the low mortality rate (~13%) of those colonized in patients without SARS-CoV-2 infection.

This highlights the importance of the ECMM/ISHAM 2020 classification for diagnosing CAPA, which includes the criteria for possible IPA. The ECMM/ISHAM 2020 consensus includes a non-bronchoscopic lavage as a new tool for the diagnosis of possible IPA. However, values for bronchoscopic lavage were based on a single study and require further validation [27]. In our center, as in many others, non-bronchoscopic lavage is not performed. Therefore, for the definition of possible IPA (only in patients with SARS-CoV-2 infection), we used respiratory samples other than BAL (sputum or TBA; not included in the ECMM/ISHAM consensus). Despite the lower specificity of these samples in patients without SARS-CoV-2 infection, we consider this justified in patients with SARS-CoV-2 infection, given the high mortality even in colonized patients.

In this series, which only includes patients with *Aspergillus* isolates in respiratory samples, there were (by definition) no possible IPA in immunocompromised patients according to EORTC criteria and very few among COPD patients. In patients with SARS-CoV-2 infection, isolates in the upper respiratory tract that did not show radiological infiltrate or showed no new radiological findings different from those presented by SARS-CoV-2 infection were considered colonized, although in many cases CT scans were not performed and many of those colonized could have been misclassified and could truly be infected. The high mortality in “colonized” patients infected by SARS-CoV-2 in this series highlights the importance of a “possible” CAPA diagnosis and the need to consider most patients with SARS-CoV-2 infection showing upper respiratory tract isolations as IPA, regardless of other diagnostic criteria, and to initiate early and effective antifungal treatment. An evaluation of a national, multi-center, prospective cohort of an enhanced testing strategy to diagnose IPA in COVID-19 intensive care patients reported a mortality rate in patients with CAPA of 57.9% (ranging from 46.7% in patients receiving appropriate antifungal therapy to 100% in patients not receiving appropriate therapy) [27].

The association between *Aspergillus* colonization and mortality has also been documented in patients with acute respiratory distress syndrome (ARDS) caused by other viruses, even with greater impact in patients with influenza. A monocenter retrospective French study comparing the incidence of ventilatory-associated pneumonia (VAP) and IPA between patients with SARS-CoV-2 (C-ARDS) and those with non-SARS-CoV-2 viral ARDS (NC-ARDS) found significantly fewer bacterial coinfections documented in C-ARDS, with fewer cases of putative aspergillosis in C-ARDS, but no difference in *Aspergillus* colonization [28]. A recent European multi-center study has compared the incidence of putative IPA in critically ill SARS-CoV-2 patients (CAPA) (n = 566) in comparison with influenza patients (IAPA) (n = 481). The incidence of putative IPA was lower in the SARS-CoV-2 pneumonia group (14, 2.5%) than in the influenza pneumonia group (29.6%), adjusted cause-specific hazard ratio. When putative IPA and *Aspergillus* respiratory tract colonization were combined, the incidence was also significantly lower in the SARS-CoV-2 group than in the influenza group (4.1% vs. 10.2%) [29].

The association between thrombocytopenia and mortality is remarkable in this series. The relationship between low platelet counts and death was more frequent and more severe in patients with SARS-CoV-2 infection, confirming a 90% mortality rate in CAPA and baseline platelet counts <100,000/µL and was also high in patients without SARS-CoV-2 infection (57.1%), being present as an independent factor in all mortality analyses. Hematological changes, including lymphopenia and thrombocytopenia, have been reported in patients with SARS-CoV-2, similar to SARS in 2003 [30,31].

Thrombocytopenia is also a common clinical manifestation associated with poor outcomes in patients with community-acquired pneumonia [32]. Recent work in 383 patients infected with SARS-CoV-2 in China confirmed that thrombocytopenia at admission (<125 × 10^9^/L) was associated with a mortality rate almost three times as high as that for those without thrombocytopenia, and this was an independent risk factor associated with in-hospital mortality in a dose-dependent manner [33]. Experimental studies have observed that thrombocytopenic mice are highly susceptible to *A. fumigatus* challenge and rapidly succumb to infection. Platelets primarily function to maintain hemostasis and lung integrity in response to exposed fungal antigens since thrombocytopenic mice exhibited severe bleeding into the airways in response to fungal challenges in the absence of overt angioinvasion [34]. Other authors have confirmed that platelets in the presence of whole plasma have the potential to play an important role in normal host defenses against invasive aspergillosis [35]. As in our series, Nouer et al. confirmed that baseline low platelet count and creatinine clearance rate predict the outcome of neutropenia-related invasive aspergillosis [36].

The availability in this series of autopsy studies confirmed IPA in four patients infected with SARS-CoV-2, three unsuspected and all four with negative serum GM. However, autopsies showed no findings of IPA in three patients who had presented pre-mortem IPA criteria. Our center pioneered performing autopsies on patients with SARS-CoV-2 infection in Spain. Our results confirm the high frequency of unsuspected IPA (3 out of 31) in patients who died of SARS-CoV-2 infection, casting doubt on the sensitivity of the available diagnostic techniques. On the other hand, the finding of negative autopsy studies in patients who had presented diagnostic criteria for IPA casts doubt on the specificity of these criteria or that the negative result is due to the effect of the treatment received. A study collecting publications with autopsies of patients with SARS-CoV-2 infection reported that autopsy-proven fungal infection occurred in only 11/550 (2%), including 8 CAPA, 2 unspecified fungal infections, and 1 disseminated mucormycosis; the authors conclude that invasive fungal infections, including CAPA, are an uncommon autopsy finding in patients with SARS-CoV-2 infection [37]. In this regard, an autopsy study conducted in Austria on patients with SARS-CoV-2 infection revealed pulmonary infarction caused by thrombosis and thromboembolism and bacterial bronchopneumonia as the most frequent cause of death and fungal pneumonia (*Aspergillus*) was found in only one case [38].

Sputum and TBA showed good sensitivities and positive predictive values in this study, especially in patients with SARS-CoV-2 infection, given their contribution to diagnosing “possible” IPA (according to the criteria used in this study). GM in BAL showed PPV >88% in all groups and, except for possible IPA in patients with SARS-CoV-2 infection, showed good negative predictive values. The low sensitivity of GM in serum is noteworthy, even in patients without SARS-CoV-2 infection, which included hematologic patients with angioinvasive forms. For diagnosis of IPA, BAL and lung biopsy samples are the choice specimens. To date, GM in BAL has been the main diagnostic test to diagnose secondary IPA in patients with severe viral infection [1,39,40]. Lateral flow techniques were not performed in this series. A multi-center retrospective study evaluated the IMMY *Aspergillus* GM lateral flow assay (LFA) with the automated reader at the 1.0 cutoff for a diagnosis of CAPA in 238 patients; combining all respiratory samples showed an AUC in ROC curves of 0.823 globally, and 0.754, 0.890, and 0.814 for BAL fluid, other no-BAL fluid and TBA, respectively. The sensitivity and specificity of serum LFA were 20% and 93%, respectively, at the 0.5 ODI cutoff [41]. Data indicate that results are similar to the enzyme immunoassay for GM, but the lateral flow test is a simple point-of-care technique and can be completed in both containment-level-3 facilities and outside specialist mycology centers, with positivity thresholds equivalent to GM testing of BAL and serum [1,2,3,4,5,6,7,8,9,10,11,12,13,14,15,16,17,18,19,20,21,22,23,24,25,26,27].

This study has several limitations. The first is the retrospective nature of the study and the single center design. The modification of the possible IPA criteria in this study—using respiratory samples other than BAL (sputum or TBA) instead of non-bronchoscopic lavage proposed in the ECMM/ISHAM 2020 consensus—is a relevant aspect; this substitution to the criteria has not been validated in other series and should be analyzed with caution. However, given the high mortality among patients with SARS-CoV-2 infection with isolation of *Aspergillus* spp. in any respiratory specimen, we believe that this experience should be considered and encourage early initiation of treatment in these patients. The inclusion (except for patients with autopsy studies) of only patients with positive *Aspergillus* isolates in respiratory samples may have excluded an undetermined number of patients who could have been diagnosed with antigenic or molecular techniques; this possibly magnifies the sensitivity of culture-based techniques in the series and reduces the sensitivity of antigenic techniques. As occurred in other series and other centers, the significant reduction of fibrobronchoscopies in the first waves of the pandemic to avoid the risk of aerosolization and transmission may have affected BAL results in the analysis. Although the population with SARS-CoV-2 infection included in this series is large, this number could have been affected by the reduction of other patients during the pandemic and the greater limitation in the performance of fibrobronchoscopies, as previously mentioned.

In summary, this study does not propose to modify the criteria for possible IPA (which requires further analysis) but to draw attention to the high mortality associated with the isolation of *Aspergillus* spp. in respiratory samples, especially in patients with SARS-CoV-2 infection, and the need for early treatment. This is more controversial in possible IPA in patients without SARS-CoV-2 infection, linked to imaging tests without microbiological findings and whose diagnostic confirmation is inferior in non-hematological patients. This large series provides an overview of IPA in hospitalized patients during a period of overlap between patients with and without SARS-CoV-2 infection and allows us to analyze some differential aspects of the clinical forms of both groups of patients. The series emphasizes the importance of baseline thrombocytopenia, especially in patients with SARS-CoV-2 infection, and renal failure as independent poor prognostic factors in IPA. Finally, despite initial limitations to FBB, this series confirms the high PPV of BAL for the diagnosis of IPA in all patients and the cost-effectiveness of *Aspergillus* isolates in upper respiratory samples in patients with SARS-CoV-2 infection that may justify early initiation of treatment.

## Figures and Tables

**Table 1 jof-09-00130-t001:** Baseline characteristics of patients with isolation of *Aspergillus* in respiratory samples.

	IPA (Proven, Probable, Possible) (n: 140)	No Aspergillosis (Colonization)	*p*
		(n: 100)	
Age (years ± SD)	66.4 (±12.6)	66.6 (±16.4)	0.89
Gender (male)	59.40%	60.60%	0.89
Tracheobronquitis	3.60%	0	0.015
Hemoptysis	11.40%	7.10%	0.37
Type of pulmonary infiltrate			
-No infiltrate	2.90%	53%	<0.001
-Bilateral, ground-glass opacities	75.00%	28.00%	<0.001
-Nodular infiltrate	9.30%	3.00%	0.067
-Caveat	3.60%	0	0.077
-Pulmonary embolism	9.30%	6.00%	0.46
-Pleural fluid	31.40%	32.50%	0.73
Diabetes	26.80%	21.20%	0.36
Diabetes (insulin required)	2.20%	1.00%	0.64
Renal insufficiency (chronic)	10.90%	7.10%	0.37
Renal insufficiency (acute or chronic)	35.00%	24.00%	0.87
Hepatic insufficiency (acute or chronic)	10.90%	9.10%	0.82
Active hematologic disorder	8.00%	0	0.003
Progenitor hematologic transplant	2.90%	1.00%	0.4
Solid organ transplant	1.40%	1.00%	1
Active Solid tumor	5.80%	5.10%	1
Active chemotherapy	3.60%	3.00%	0.84
Corticoid therapy	10.1	3	0.04
Corticoid therapy (>15 mg prednisone/d)	4.30%	0	0.04
Therapy including biologics	2.90%	1.00%	0.4
Neutrophils <100/µL (last 3 months)	1.40%	0	0.51
Neutrophils count (±SD)	8380 (±6100)	8076 (±5400)	0.7
Lymphocytes count (±SD)	1304 (±1472)	1492 (±1347)	0.33
Platelets count (±SD)	223.6 × 10^3^ (±106.2 ×10^3^)	230.7 × 10^3^ (±113.7 × 10^3^)	0.63
D-Dimer count (±SD)	6611 (±13,049)	2540 (±2834)	0.17
IL-6 (±SD)	227.5 (±706.5)	826.5 (±1850.5)	<0.001
CRP (±SD)	85.9 (±88.0)	119.6 (±119.0)	0.03
Previous aspergillosis	1.40%	3.00%	0.65
Oxygen any requirements	88.60%	62%	<0.001
Oxygen high requirements	68.60%	45.00%	<0.001
ICU admission	63.60%	42.00%	0.001
Mechanical ventilation	58.60%	39.00%	<0.001
Inotropic therapy	50.70%	33.00%	0.008
Pronation required	48.60%	21.00%	<0.001
ECMO	2.10%	1.00%	0.17
Corticoid therapy previous (any dose)	83.60%	41.40%	<0.001
Corticoid therapy previous (>250 mg/d)	3.60%	1.00%	0.4
Bacterial infection	87.10%	65.00%	<0.001
CMV infection	43.50%	40.00%	1
CMV disease	12.90%	25.00%	0.28
Hospitalization total days	30.5 d (±25.4)	22.5 d (±33.3)	0.04
ICU total days	16.5 d (±19.7)	10.8 d (±17.8)	0.029
Dead	37.10%	34.00%	0.61
SARS-CoV-2 infection	77.10%	24.00%	<0.001
Autopsy available	5.00%	24%	<0.001
Autopsy with IPA confirmation	2.90%	-	-

ECMO: extracorporeal membrane oxygenation. IL-6: Interleukin-6. CMV: Cytomegalovirus. CRP: C-reactive protein. ICU: Intensive care unit.

**Table 2 jof-09-00130-t002:** Analysis of mortality. Uni- and multivariate analysis including all patients.

	Univariate Analysis		Multivariate Analysis		
	Global Mortality	*p*	OR	IC95%	*p*
	86/240 (35.8%)				
Age >65 y-o	39.90%	0.29	2.69	1.29–5.57	0.008
Gender (male)	44.40%	0.002	-	-	-
Proven, probable or possible IPA	37.10%	0.68	-	-	-
Proven, or probable IPA	43.50%	0.23	-	-	-
Type of pulmonary infiltrate			-	-	-
-No infiltrate	18.80%	0.001	-	-	-
-Nodular or bilateral, ground-glass opacities	48.10%	<0.001	-	-	-
-Pleural fluid	54.90%	0.001	-	-	-
Renal insufficiency (acute or chronic)	56.20%	<0.001	2.58	1.30–5.08	0.006
Neutrophils <100/µL	50.00%	0.001	-	-	-
Lymphocytes <1000/µL	48.50%	0.035	-	-	-
Platelets count <100.000/µL	73.90%	<0.001	6.41	2.07–19.8	0.001
CRP (±SD)	117 (±106)	0.018	-	-	-
Oxygen high requirements	53.20%	<0.001	-	-	-
ICU admission	51.90%	<0.001	-	-	-
Mechanical ventilation	53.70%	<0.001	-	-	-
Inotropic therapy	61.50%	<0.001	5.32	2.52–11.19	<0.001
Pronation required	58.40%	<0.001	-	-	-
Corticoid therapy previous (any dose)	44.90%	<0.001	-	-	-
Previous corticoids	16.2 (±9.7)	0.004	-	-	-
Bacterial infection	42.20%	<0.001	-	-	-
CMV infection	62.90%	<0.079	-	-	-
Hospitalization total days	31 (±23)	0.09	-	-	-
ICU total days	20 (±18)	0.001	-	-	-
SARS-CoV-2 infection	51.50%	<0.001	3.14	1.40–7.04	0.005

ECMO: extracorporeal membrane oxygenation. IL-6: Interleukin-6. CMV: Cytomegalovirus. CRP: C-reactive protein. ICU: Intensive care unit.

**Table 3 jof-09-00130-t003:** Baseline characteristics of patients with IPA (proven, probable, or possible).

	SARS-CoV-2(+) (n: 108)	SARS-CoV-2(-) (n: 32)	*p*
Age (years ± SD)	65.5 (±12.1)	69.4 (±14.2)	0.18
Gender (male)	64.80%	40.00%	0.02
Tracheobronchitis	10.7% (3/28)	11.1% (2/18)	0.89
Hemoptysis	11.10%	12.50%	0.76
Proven or probable invasive aspergillosis	22.20%	68.80%	<0.001
Type of pulmonary infiltrate			
-No infiltrate	1.90%	6.20%	0.22
-Bilateral, ground-glass opacities	81.50%	53.10%	0.002
-Nodular infiltrate	2.80%	31.20%	<0.001
-Caveat	1.90%	9.40%	0.08
-Pulmonary embolism	12.00%	0	0.04
-Pleural fluid	34.30%	21.90%	0.34
Diabetes	27.80%	23.30%	0.81
Diabetes (insulin required)	2.80%	0	1
Renal insufficiency (chronic)	11.10%	10.00%	1
Renal insufficiency (acute or chronic)	37.00%	28.10%	0.4
Hepatic insufficiency (chronic)	3.70%	3.30%	1
Hepatic insufficiency (acute or chronic)	13.00%	3.30%	0.19
Active hematologic disorder	4.60%	20.00%	0.014
Progenitor hematologic transplant	0.90%	10%	0.032
Solid organ transplant	0.90%	3.30%	0.38
Active chemotherapy (solid tumor)	1.90%	20.00%	0.001
Corticoid therapy	7.40%	20.00%	0.08
Corticoid therapy (>15 mg prednisone/d)	3.70%	6.70%	0.61
Therapy including biologics	1.90%	6.70%	0.2
Neutrophils <100/µL (last 3 months)	0.90%	3.10%	0.4
Neutrophils count (±SD)	8800 (±6200)	6500 (±5400)	0.06
Lymphocytes count (±SD)	1094 (±912)	2061 (±2500)	0.049
Platelets count (±SD)	231 × 10^3^ (±104 × 10^3^)	197 × 10^3^ (±108 × 10^3^)	0.13
D-Dimer count (±SD)	6600 (±13,100)	-	-
IL-6 (±SD)	227 (±740)	-	-
CRP (±SD)	88.3 (±87.2)	75.8 (±5400)	0.53
Previous aspergillosis	0	6.20%	0.05
Oxygen any requirements	98.10%	56.20%	<0.001
Oxygen high requirements	85.20%	12.50%	<0.001
ICU admission	78.70%	12.50%	<0.001
Mechanical ventilation	73.10%	9.40%	<0.001
Inotropic therapy	63.00%	9.40%	<0.001
Pronation required	62.00%	3.10%	<0.001
ECMO	0.90%	0	0.54
ICU days	25 (±19)	26 (±17)	0.91
Previous corticoid therapy (any dose)	95.40%	43.80%	<0.001
Previous corticoid therapy (>250 mg/d)	4.60%	0	0.59
Bacterial infection	90.70%	75.00%	0.03
CMV infection	38.20%	87.70%	0.03
CMV disease	10.90%	28.60%	0.22
Remdesivir therapy	23.40%	0	0.001
Tocilizumab therapy	29.00%	0	<0.001
Initial Aspergillosis therapy			
-Voriconazole	23.10%	25.00%	0.89
-Isavuconazole	24.10%	6.20%	0.03
-Liposomal Amphotericin B	7.40%	6.20%	0.82
Hospitalization total days	35 (±25)	14 (±17)	<0.001
ICU total days	20 (±20)	3.6 (±11)	<0.001
Dead	40.70%	25.00%	0.14
Related mortality	17.60%	9.40%	0.16

ECMO: extracorporeal membrane oxygenation. IL-6: Interleukin-6. CMV: Cytomegalovirus. RCP: reactive C Protein. ICU: Intensive care unit.

**Table 4 jof-09-00130-t004:** Analysis of mortality. Uni- and multivariate analysis. Patients with IPA (proven, probable or possible).

	Univariate Analysis		Multivariate Analysis		
	Global Mortality	*p*	OR	IC95%	*p*
	46/140 (32.9%)				
Age > 65 y-o	40.20%	0.048	-	-	-
Gender (male)	46.30%	0.013	-	-	-
Proven or probable invasive aspergillosis	43.50%	0.35	-	-	-
Pleural fluid	52.30%	0.015	-	-	-
Renal insufficiency (acute or chronic)	61.20%	0.001	4.93	2.12–11.5	<0.001
Neutrophils count (±SD)	10,492 (±8100)	0.002	-	-	-
Platelets < 100,000/µL	77.80%	<0.001	37.91	2.16–29.0	0.002
Oxygen any requirements	41.90%	0.001	-	-	-
Oxygen high requirements	46.90%	<0.001	-	-	-
ICU admission	46.10%	0.004	-	-	-
Mechanical ventilation	46.30%	0.008	-	-	-
Inotropic therapy	54.90%	<0.001	4.89	2.07–11.5	<0.001
SARS-CoV-2 infection	40.70%	0.14	-	-	-

ICU: Intensive care unit.

**Table 5 jof-09-00130-t005:** Characteristics of patients with autopsy study available.

	Mold Infection Confirmed in Autopsy (n: 4)	Mold Infection Not Confirmed in Autopsy (n: 27)
SARS-CoV-2 infection	4/4 (100%)	27/27 (100%)
Proven, probable or possible IPA	4/4 (100%)	3/27 (11.1%)
Proven or probable IPA	4/4 (100%)	2/27 (7.4%)
*Aspergillus* isolated in sputum	1/4(25%)	4/27 (14.8%)
*Aspergillus* isolated in tracheo-bronchial aspirate	0/4 (0%)	3/27 (11.1%)
*Aspergillus* isolated in BAL (all patients)	0/4 (0%)	1/27 (3.7%)
*Aspergillus* isolated in BAL (if available)	0/2 (0%)	1/4(25%)
Hyphae were observed in BAL	0/4 (0%)	1/27 (3.7%)
Positive GM in BAL (all patients)	1/4(25%)	1/27 (3.7%)
Positive GM in serum (all patients)	0/4 (0%)	0/27 (0%)
Type of pulmonary infiltrate		
-No infiltrate	0/4 (0%)	0/27 (0%)
-Nodular or bilateral, ground-glass opacities	1/4 (25%)	23/27 (85.2%)
-Caveat	0/4 (0%)	0/27 (0%)
-Pulmonary embolism	3/4 (75%)	4/27 (14.8%)
-Pleural fluid	2/4 (50%)	11/27 (40.7%)

IPA: invasive pulmonary aspergillosis. GM: galactomannan. BAL: bronchoalveolar lavage.

**Table 6 jof-09-00130-t006:** Performance of diagnostic approach.

	Sensitivity	Specificity	Positive Predictive Value	Negative Predictive Value
PPP IPA in patients with SARS-CoV-2 infection (IPA: 108, no infection: 24)				
Sputum (n: 39) (positive: 29, negative: 10)	29/36 (80.5%)	3/3 (100%)	29/29 (100%)	3/10 (30.0%)
TBA (n: 84) (positive: 76, negative: 27)	76/84 (90.4%)	19/19 (100%)	76/76 (100%)	19/27 (70.4%)
BAL (n: 43) (positive: 16, negative: 27)	16/40 (40.0%)	3/3 (100%)	16/16 (100%)	3/27 (11.1%)
Hyphae in BAL (n: 18) (positive: 3, negative: 15)	3/18 (16.7%)	0/0	3/3 (100%)	0/15 (0)
GM in BAL (n: 39) (positive: 18, negative: 21)	18/37 (48.6%)	2/2 (100%)	18/18 (100%)	2/21 (9.5%)
GM in serum (n: 84) (positive: 8, negative: 76)	8/72 (11.1%)	12/12 (100%)	8/8 (100%)	12/76 (15.8%)
PP IPA in patients with SARS-CoV-2 infection (IPA: 24, no infection: 108)				
Sputum (n: 39) (positive: 29, negative: 10)	2/4 (50%)	8/35 (22.8%)	2/29 (6.9%)	8/10 (80%)
TBA (n: 84) (positive: 76, negative: 27)	19/24 (79.2%)	22/79 (27.8%)	19/76 (25.0%)	22/27 (81.5%)
BAL (n: 43) (positive: 16, negative: 27)	13/20 (65.0%)	20/23 (86.9%)	13/16 (81.2%)	20/27 (74.1%)
Hyphae in BAL (n: 18) (positive: 3, negative: 15)	2/10 (20.0%)	7/8 (87.5%)	2/3 (66.7%)	7/15 (46.5%)
GM in BAL (n: 39) (positive: 18, negative: 21)	16/21 (76.2%)	16/18 (88.9%)	16/18 (88,9%)	16/21 (76.2%)
GM in serum (n: 84) (positive: 8, negative: 76)	4/21 (19.0%)	59/63 (93.6%)	4/8 (50%)	59/76 (77.6%)
PPP IPA in patients without SARS-CoV-2 (IPA: 32, no infection: 76)				
Sputum (n: 70) (positive: 62, negative: 8)	16/20 (80%)	4/50 (8.0%)	16/62 (25.8%)	4/8 (50%)
TBA(n: 56) (positive: 44, negative: 12)	13/20 (65%)	5/36 (13.9%)	13/44 (29.5%)	5/12 (41.7%)
BAL (n: 25) (positive: 13, negative: 12)	11/17 (64.7%)	6/8 (75.0%)	11/13 (84.6%)	6/12 (50%)
Hyphae in BAL (n: 4) (positive: 2, negative: 2)	2/2 (100%)	2/2 (100%)	2/2 (100%)	2/2 (100%)
GM in BAL (n: 17) (positive: 4, negative: 13)	4/11 (36.4%)	6/6 (100%)	4/4 (100%)	6/13 (46.2%)
GM in serum (n: 19) (positive: 2, negative: 17)	2/9 (22.2%)	10/10 (100%)	2/2 (100%)	10/17 (58.8%)
PP IPA in patients without SARS-CoV-2 (IPA: 22, no infection: 86)				
Sputum (n: 70) (positive: 62, negative: 8)	12/13 (92.3%)	7/57 (12.2%)	12/62 (19.4%)	7/8 (87.5%)
TBA (n: 56) (positive: 44, negative: 12)	8/13 (61.5%)	7/43 (16.2%)	8/44 (18.2%)	7/12 (58.3%)
BAL (n: 25) (positive: 13, negative: 12)	9/13 (69.2%)	8/12 (66.7%)	9/13 (69.2%)	8/12 (66.7%)
Hyphae in BAL (n: 4) (positive: 2, negative: 2)	1/1 (100%)	2/3 (66.7%)	1/2 (50%)	2/2 (100%)
GM in BAL (n: 17) (positive: 4, negative: 13)	4/8 (50%)	9/9 (100%)	4/4 (100%)	9/13 (69.2%)
GM in serum (n: 19) (positive: 2, negative: 17)	2/8 (25%)	11/11 (100%)	2/2 (100%)	11/17 (64.7%)

IPA: invasive pulmonary aspergillosis. GM: galactomannan. BAL: bronchoalveolar lavage. TBA: tracheo-bronchial aspirate.

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
