# Peer review of "Invasive Pulmonary Aspergillosis in Patients with and without SARS-CoV-2 Infection"

_jof, 2023, doi:10.3390/jof9020130_

Round 1

Reviewer 1 Report (Previous Reviewer 2)

This is a much improved version of the manuscript. I still recommend to change the title ''Invasive pulmonary aspergillosis: The role of Aspergillus colonization in patients with and without SARS-CoV-2 infection''.   Do the authors want to empasize on invasive pulmonary aspergillosis or Aspergillus colonization or  the role of  SARS-CoV-2 infection?

Author Response

Thank you for your interest in the manuscript.

Following your suggestion we have modified the title as follows: "Invasive pulmonary aspergillosis in patients with and without SARS-CoV-2 infection"

English language and style have been revised

In introduction section, lines 56 to 61 have been deleted. Line 78 has been modified: A diagnostic, clinical, and prognostic characteristic of patients.

Line 103 has been modified: "Patients <18 years old, or patients with an absence of Aspergillus spp isolation in a respiratory specimen"

In methods line 111 has been modified "ELISA) (Platelia™ Aspergillus ELISA, Bio-Rad, Marnes-la-Coquette, France)". All the different diagnosis criteria of IPA used in the manuscript have been revised: the recently modified EORTC/MSG criteria, in immunocompromised patients; the AspICU definitions for probable/putative IPA, in ICU patients; the Bulpa criteria, in patients with chronic obstructive pulmonary disorder (COPD) and the modified ECMM/ISHAM 2020 consensus criteria in SARS-CoV-2 infection

In results section all tables and results mentioned in the text have been checked.

The discussion has been reviewed in detail. The content of the lines 377-380 has been strengthened In patients with SARS-CoV-2 infection, isolates in the upper respiratory tract that did not show radiological infiltrate o showed no new radiological findings different from those presented by SARS-CoV-2 infection were considered colonized,

A new version of the manuscript is included, incorporating the changes made in yellow. 

Reviewer 2 Report (Previous Reviewer 3)

Thank you!

Author Response

Thank you very much for your review. Following your recommendations, the following revisions have been made to the manuscript: the introduction background, a better description of the design and methods and better presentation of results and conclusions

In introduction section, lines 56 to 61 have been deleted. Line 78 has been modified: A diagnostic, clinical, and prognostic characteristic of patients.

Line 103 has been modified: "Patients <18 years old, or patients with an absence of Aspergillus spp isolation in a respiratory specimen"

In methods line 111 has been modified "ELISA) (Platelia™ Aspergillus ELISA, Bio-Rad, Marnes-la-Coquette, France)". All the different diagnosis criteria of IPA used in the manuscript have been revised: the recently modified EORTC/MSG criteria, in immunocompromised patients; the AspICU definitions for probable/putative IPA, in ICU patients; the Bulpa criteria, in patients with chronic obstructive pulmonary disorder (COPD) and the modified ECMM/ISHAM 2020 consensus criteria in SARS-CoV-2 infection

In results section all tables and results mentioned in the text have been checked.

The discussion has been reviewed in detail. The content of the lines 377-380 has been strengthened In patients with SARS-CoV-2 infection, isolates in the upper respiratory tract that did not show radiological infiltrate o showed no new radiological findings different from those presented by SARS-CoV-2 infection were considered colonized,

A new version of the manuscript is included, incorporating the changes made in yellow.

This manuscript is a resubmission of an earlier submission. The following is a list of the peer review reports and author responses from that submission.

Round 1

Reviewer 1 Report

In this single-center study, the authors describe a cohort of hospitalized patients with isolation of Aspergillus spp in respiratory cultures. They classify the patients into the groups invasive pulmonary aspergillosis (IPA) and colonization, respectively, and describe risk factors for mortality and the role of SARS-CoV-2. In addition, they present postmortem data analyzing the presence of IPA in diseased patients with COVID. Although the topic is relevant and the authors present a lot of interesting data, the manuscript is somewhat difficult to follow and would benefit from more editing.

Major comments:

-       My biggest concern regarding this manuscript is that the aim of the study is a bit unclear (both evident in the title and in the manuscript text) and the large amount of data makes it therefore difficult to follow and to evaluate. Is the aim to study risk factors for mortality in hospitalized patients with growth of aspergillus in the airways? Or is it to compare clinical characteristics of IPA-patients with and without COVID? Or is it to study the risk of IPA in COVID-patients? Or is it to evaluate the performance of different diagnostic tools relating to IPA and CAPA? It would be easier for the reader if the authors clearly state the aim of the study in the abstract and introduction, and then focus the results and discussion around this specific research question.

-       Exclusion critera, lines 93-94: It is stated that absence of SARS-CoV-2 infection is an exclusion criterium. Is that correct?

-       Diagnostic criteria of possible CAPA, lines 117-122: What kind of radiological findings were required for the classification possible CAPA? Given that COVID also causes infiltrates on chest x-ray and CT scans, did you differentiate between COVID-related and aspergillosis-related infiltrates? 

-       Table 1: How many patients in the IPA-group fulfilled the criteria for proven, probable and possible IPA, respectively? If I read the manuscript correctly, as many as 96 of the 140 IPA-patients (87%) had a possible IPA, of which 84 had COVID. Relating to the point above about criteria for possible CAPA, and the fact that 75% of the patients in the IPA-group had non-specific “bilateral alveolo-interstitial infiltrates”, I wonder how well possible CAPA fits into the IPA-group? What would the results of Table 1 look like if only proven and probable IPA (PP-IPA) were compared to colonization?

Minor comments:

-       Table 1: I think the data on the last 9 rows have been shifted up one step.

-       Table 2 and 3: PCR should perhaps be CRP?

-       Table 2: How do you define “Oxygen high requirements”?

-       Table 3: How do you define CMV infection and disease, respectively?

-       Table 3: What is “related mortality” and how is it defined?

Author Response

Thank you very much for your attention and interest in our manuscript.
This new version is uploaded to the digital platform. The following modifications are detailed point by point and highlighted in yellow in the text.

Rev 1
Query
Comments and Suggestions for Authors
In this single-center study, the authors describe a cohort of hospitalized patients with isolation of Aspergillus spp in respiratory cultures. They classify the patients into the groups invasive pulmonary aspergillosis (IPA) and colonization, respectively, and describe risk factors for mortality and the role of SARS-CoV-2. In addition, they present postmortem data analyzing the presence of IPA in diseased patients with COVID. Although the topic is relevant and the authors present a lot of interesting data, the manuscript is somewhat difficult to follow and would benefit from more editing.

Major comments:
-       My biggest concern regarding this manuscript is that the aim of the study is a bit unclear (both evident in the title and in the manuscript text) and the large amount of data makes it therefore difficult to follow and to evaluate. Is the aim to study risk factors for mortality in hospitalized patients with growth of aspergillus in the airways? Or is it to compare clinical characteristics of IPA-patients with and without COVID? Or is it to study the risk of IPA in COVID-patients? Or is it to evaluate the performance of different diagnostic tools relating to IPA and CAPA? It would be easier for the reader if the authors clearly state the aim of the study in the abstract and introduction, and then focus the results and discussion around this specific research question.
Answer                                                                                                               The most important objective is to highlight the relevance of Aspergillus spp colonization in critically ill patient, and especially in the patients with SARS-COV-2 infection. In addition, the study takes the opportunity to compare two populations: with and without SARS-COV-2 infection. Analyzes the different criteria and the cost-effectiveness of diagnostic techniques. It also provides necropsy data. These are different objectives but we believe that they are all related to each other. We believe that the separate publication of these objectives would not maintain the global vision of the study.
Query
-       Exclusion critera, lines 93-94: It is stated that absence of SARS-CoV-2 infection is an exclusion criterium. Is that correct?
Answer
This is a mistake. Absence of SARS-CoV-2 infection is not an exclusion criterion. This sentence has been deleted

Query
Diagnostic criteria of possible CAPA, lines 117-122: What kind of radiological findings were required for the classification possible CAPA? Given that COVID also causes infiltrates on chest x-ray and CT scans, did you differentiate between COVID-related and aspergillosis-related infiltrates?
Answer
These are the same radiological criteria applied for probable IPA, set out in the previous paragraph: require a pulmonary infiltrate or nodules, preferably documented by chest CT, or cavitating infiltrate.

Query
Table 1: How many patients in the IPA-group fulfilled the criteria for proven, probable and possible IPA, respectively? If I read the manuscript correctly, as many as 96 of the 140 IPA-patients (87%) had a possible IPA, of which 84 had COVID. Relating to the point above about criteria for possible CAPA, and the fact that 75% of the patients in the IPA-group had non-specific “bilateral alveolo-interstitial infiltrates”, I wonder how well possible CAPA fits into the IPA-group? What would the results of Table 1 look like if only proven and probable IPA (PP-IPA) were compared to colonization?
Answer
As mentioned in Table 3, 22.2% of the 108 SARS-COV-2 patients with IPA were PP-IPA. Then the remaining 77.8% (72) were possible IPA in this group. Table 1 shows that 75% of the PPP-IPA showed alveolo-interstitial infiltrate (this infiltrate in possible IPA was similar, 78%) confirmed mostly on CT. The 28% of alveolo-interstitial infiltrates in colonized patients (table 1) referred to X-ray because they were not confirmed on CT. As mentioned in the text (rows 264-269) the analysis in PP-IPA and PPP-IPA was similar:  "Comparison between patients with or without SARS-COV-2 infection (n: 24 vs n: 22) in an analysis restricted only to the population PP-IPA also showed similar results. Multivariate analysis was also concordant in this group and showed the same factors independently associated with higher mortality"

Query
Minor comments:
Table 1: I think the data on the last 9 rows have been shifted up one step.
Answer
The decoupling of rows in tables has been corrected.

Query
Table 2 and 3: PCR should perhaps be CRP?
Answer
Yes, it is true. It has been corrected

Query
Table 2: How do you define “Oxygen high requirements”?
Answer
high-flow oxygen or use of a reservoir.

Query
Table 3: How do you define CMV infection and disease, respectively?
Answer
CMV infection is viral load in the absence of symptoms or signs of disease.

Query
Table 3: What is “related mortality” and how is it defined?
Answer
related mortality was considered when no other causes were observed.

Reviewer 2 Report

In the manuscript entitled ‘’Invasive pulmonary aspergillosis: The role of Aspergillus colonization in patients with and without SARS-CoV-2 infection’’, the authors provide a detailed analysis  of the epidemiological, diagnostic, and prognostic differences of IPA in patients with and without SARS-CoV-2 infection. They have shown that the isolation of Aspergillus spp in respiratory samples, with or without IPA criteria (colonization), is associated with high morbidity and mortality in a series of  hospitalized patients and mainly in ICU admits  of a tertiary hospital. The series emphasizes the importance of baseline thrombocytopenia, especially in COVID+, and renal failure as independent poor prognostic factors in IPA. Finally, despite initial limitations to FBB in COVID+ patients, this series confirms the high PPV of BAL for the diagnosis of IPA in all patients and the cost-effectiveness of Aspergillus isolates in upper respiratory samples in patients with SARS-CoV-2  infection that may justify early initiation of treatment.

This is a well-written manuscript. The methodology used and the statistical analysis are adequate and the results are clearly presented. The discussion section is consistent with the findings of the study.Minor revisions are needed.

-      Title: The authors may change the title so as to emphasize the main finding of the paper, which is the high morbitity and mortality of patients with Aspergillus spp.

-      Line 27: please remove the underline format in the text

-      Line 40: please replace not- with non-

-      Line 55: Please replace (Influenza Associated Aspergillosis) IAPA with Influenza Associated Aspergillosis (IAPA).

-      lines 93-94 may be moved to the previous paragraph (2.3), and rename the paragraph as 2.3 Inclusion/Exclusion criteria. Paragraph 2.4 (lines 95-106) may be renamed as 2.4 Microbiological methods.

-      Line 129: please define COPD

-      Line 170: Please replace All patients with  Characteristics of patients with isolation of Aspergillus spp

-      Table 1: please check  the format style of the table . The percentages 87.1%, 65% p<000.1 correspond to the row  of Bacterial infection.

-      Lines 189-190: I would suggest to perform a A Chi-Square Test of Independence so as to determine whether or not there is a significant association between these two categorical variables (IPA/Colonization vs SARS-CoV-2 positive/negative).

-      Line 214: please replace In with in.

-      Line 235: please correct analyzes with summarizes

-      Table 6: please check the numbers COVID+. IPA (proven, probable or possi-ble) (IPA: 108, no infection: 24) and COVID+. IPA (proven, probable) (IPA: 24, no infection: 108)

-      Line 362: please check the font size.

Author Response

Thank you very much for your attention and interest in our manuscript.
This new version is uploaded to the digital platform. The following modifications are detailed point by point and highlighted in yellow in the text.

Reviewer: 2

Query
In the manuscript entitled ‘’Invasive pulmonary aspergillosis: The role of Aspergillus colonization in patients with and without SARS-CoV-2 infection’’, the authors provide a detailed analysis  of the epidemiological, diagnostic, and prognostic differences of IPA in patients with and without SARS-CoV-2 infection. They have shown that the isolation of Aspergillus spp in respiratory samples, with or without IPA criteria (colonization), is associated with high morbidity and mortality in a series of  hospitalized patients and mainly in ICU admits  of a tertiary hospital. The series emphasizes the importance of baseline thrombocytopenia, especially in COVID+, and renal failure as independent poor prognostic factors in IPA. Finally, despite initial limitations to FBB in COVID+ patients, this series confirms the high PPV of BAL for the diagnosis of IPA in all patients and the cost-effectiveness of Aspergillus isolates in upper respiratory samples in patients with SARS-CoV-2  infection that may justify early initiation of treatment.
This is a well-written manuscript. The methodology used and the statistical analysis are adequate and the results are clearly presented. The discussion section is consistent with the findings of the study.Minor revisions are needed.
-      Title: The authors may change the title so as to emphasize the main finding of the paper, which is the high morbitity and mortality of patients with Aspergillus spp.
Answer
We agree, but in addition to the high morbidity-mortality of aspergillosis, the paper also includes other aspects. We believe that the current title encompasses all of them.

Query
-      Line 27: please remove the underline format in the text
-      Line 40: please replace not- with non-
Answer
OK.

Query
Line 55: Please replace (Influenza Associated Aspergillosis) IAPA with Influenza Associated Aspergillosis (IAPA).

Answer
OK.

Query
lines 93-94 may be moved to the previous paragraph (2.3), and rename the paragraph as 2.3 Inclusion/Exclusion criteria. Paragraph 2.4 (lines 95-106) may be renamed as 2.4 Microbiological methods.

Answer
ok.
The paragraphs have been reordered

Query
  Line 129: please define COPD

Answer
Yes. It has been defined

Query
-      Line 170: Please replace All patients with  Characteristics of patients with isolation of Aspergillus spp
Answer
Yes. It has been changed

Query
Table 1: please check  the format style of the table . The percentages 87.1%, 65% p<000.1 correspond to the row  of Bacterial infection.
Answer
The decoupling of rows in tables has been corrected.

Query
Lines 189-190: I would suggest to perform a A Chi-Square Test of Independence so as to determine whether or not there is a significant association between these two categorical variables (IPA/Colonization vs SARS-CoV-2 positive/negative).
Answer
Yes. The distribution of colonized patients with and without SARS-COV-2 infection is significant, as discussed below.

Query
-      Line 214: please replace In with in.
-      Line 235: please correct analyzes with summarizes
Answer
OK

Query
Table 6: please check the numbers COVID+. IPA (proven, probable or possi-ble) (IPA: 108, no infection: 24) and COVID+. IPA (proven, probable) (IPA: 24, no infection: 108) -      Line 362: please check the font size.
Answer
Yes. It has been checked

Reviewer 3 Report

Fortún et al. present a retrospective single-center study comparing cases of Aspergillus colonization (n = 100) and infection (n = 140) according to the EORTC/MSG, the CAPA, the aspICU, and the Bulpa criteria. It is demonstrated that mortality rates are comparable, which questions the ability of the applied definitions to discriminate between infection and colonization and which highlights the importance of the recovery of Aspergillus from respiratory samples. The identification of the significance of platelet counts and the combination of SARS-CoV-2 infection and colonization are notable findings.

The results of this study are of interest for the scientific and the medical community. However, the manuscript needs to be revised before being suitable for publication in JoF.

Major comments

The paper is very unpleasant to read. This is due to numerous superficialities, sloppy formulations and trivial errors that should not occur in a professional scientific publication. For instance:

1.            COVID-19 is neither the abbreviation for “coronavirus 2019” (Ll.45-46) nor for “SARS-Cov2 disease” (L.189).

2.            It is COVID-19 but not COVID (Table 1, L.121, L.192, L.194, L.197,…)

3.            It is SARS-CoV-2 but not SARS-Cov2 (L.188).

4.            It is Aspergillus (in italic) but not aspergillus (L.66).

5.            IAPA is Influenza associated pulmonary aspergillosis but not Influenza associated aspergillosis (L.55).

6.            CAPA is not the abbreviation for “pulmonary aspergillosis associated with SARS-CoV-2 infection”.

7.            It is CRP not PCR or RCP (Table 2, Table 3, Ll.247-248).

8.            Identifying information about diagnostic tools should be consistent: “(Thermofisher, USA)” vs. “(PlateliaTM Aspergillus; Bio-Rad Laboratories)” vs. “(Bruker)”

9.            Please adhere either to British or to American English.

10.          Several times, there is a sudden change to other fonts or font sizes.

Several comparisons between the IPA and the colonization cohort should be considered with caution, since there is a severe risk of bias. For instance, the IPA cohort was described to feature radiological findings more frequently compared to the colonization cohort (Ll.177-179, Table 1). This is misleading since the IPA cohort is (at least for the majority [?] of its cases) defined by the presence of those radiological findings. In other words: many of the respective cases of the colonization cohort are only included in this cohort because they do not have those radiological findings and would become part of the IPA cohort as soon as the had those radiological findings (EORTC/MSG criteria). Same probably for some clinical conditions / underlying diseases that are included in the host facto criteria of the EORTC7MSG definitions, e.g., hematologic disorder, organ transplant, corticosteroid therapy,… (Ll.179-180, Table 1).

Why do we need paragraph 2.5? It explains the diagnostic criteria (EORTC/MEG, CAPA, AspICU, Bulpa), which are perfectly depicted in the respective publications cited by the authors. Some information are even misleading or at least irritating for the reader: what does the asterisk indicate (L.120)? Why are case definitions explained, which are stated to be not relevant for this study (“not available in this study” / “not performed in this study”) (Ll.115-116, Ll.118-119).

The impact of the modification of the CAPA criteria remains unclear. How many cases were classified as CAPA, which otherwise would be not, based on this modification? What would be the result of this manuscript’s key messages when those cases would have to be excluded from the PPP group? Would we still get the overall same results?

Minor comments

L.35: “confirmed independent variables” please rephrase

L.39: importance

Ll.48-54: The presented data should be summarized. 202?

L.155: PPP-IPA: proven, probable, possible – how about putative?

L.186; how about acute renal insufficiency? Last line of table?

L.201: “severity of colonized patients” please rephrase

L.214-216: please clarify the sentence.

L.218-219: mortality [was] observed?

Ll271-272: please rephrase

L.279: proven instead of proved

Ll.302-303: please add a verb.

Ll.303-304: how about reference 29?

Ll.313-315: “the prognosis is worse in infected patients” vs. “the similar results (even higher mortality in colonized patients” seems to be contradictive.

Ll.414-431: another limitation: single center design

Author Response

Thank you very much for your attention and interest in our manuscript.
This new version is uploaded to the digital platform. The following modifications are detailed point by point and highlighted in yellow in the text.

Reviewer: 3
Fortún et al. present a retrospective single-center study comparing cases of Aspergillus colonization (n = 100) and infection (n = 140) according to the EORTC/MSG, the CAPA, the aspICU, and the Bulpa criteria. It is demonstrated that mortality rates are comparable, which questions the ability of the applied definitions to discriminate between infection and colonization and which highlights the importance of the recovery of Aspergillus from respiratory samples. The identification of the significance of platelet counts and the combination of SARS-CoV-2 infection and colonization are notable findings.
The results of this study are of interest for the scientific and the medical community. However, the manuscript needs to be revised before being suitable for publication in JoF.

Query
Major comments
The paper is very unpleasant to read. This is due to numerous superficialities, sloppy formulations and trivial errors that should not occur in a professional scientific publication. For instance:
Answer
The entire manuscript has been revised and proofread by professional Native American translators (American Manuscripts Editors).

Query
1.            COVID-19 is neither the abbreviation for “coronavirus 2019” (Ll.45-46) nor for “SARS-Cov2 disease” (L.189).
2.            It is COVID-19 but not COVID (Table 1, L.121, L.192, L.194, L.197,…)
3.            It is SARS-CoV-2 but not SARS-Cov2 (L.188).
4.            It is Aspergillus (in italic) but not aspergillus (L.66).
5.            IAPA is Influenza associated pulmonary aspergillosis but not Influenza associated aspergillosis (L.55).
6.            CAPA is not the abbreviation for “pulmonary aspergillosis associated with SARS-CoV-2 infection”.
Answer
The following has been standardized and is used throughout the manuscript: SARS-COV-2 or SARS-COV-2 infection.

Query
7.            It is CRP not PCR or RCP (Table 2, Table 3, Ll.247-248).
Answer
Yes. It has been changed

Query
8.            Identifying information about diagnostic tools should be consistent: “(Thermofisher, USA)” vs. “(PlateliaTM Aspergillus; Bio-Rad Laboratories)” vs. “(Bruker)”
Answer
The definition and reference of microbiological tests has been extended: QuantStudio S5 Real-time PCR system; ThermoFisher, USA; Platelia™ Aspergillus; Bio-Rad Laboratories; Bruker Biotyper; Bruker Daltonics, Germany

Query
9.            Please adhere either to British or to American English.
Answer
The entire manuscript has been revised and proofread by professional Native American translators (American Manuscripts Editors).

Query
10.          Several times, there is a sudden change to other fonts or font sizes.
Answer
It has been corrected

Query
Several comparisons between the IPA and the colonization cohort should be considered with caution, since there is a severe risk of bias. For instance, the IPA cohort was described to feature radiological findings more frequently compared to the colonization cohort (Ll.177-179, Table 1). This is misleading since the IPA cohort is (at least for the majority [?] of its cases) defined by the presence of those radiological findings. In other words: many of the respective cases of the colonization cohort are only included in this cohort because they do not have those radiological findings and would become part of the IPA cohort as soon as the had those radiological findings (EORTC/MSG criteria). Same probably for some clinical conditions / underlying diseases that are included in the host facto criteria of the EORTC7MSG definitions, e.g., hematologic disorder, organ transplant, corticosteroid therapy,… (Ll.179-180, Table 1).

Answer
Yes. This aspect has also been discussed with previous reviewers. Limitations between possible and colonized cases, especially in SARS-COV-2 infected patients. In any case, as stated in the document, the analysis restricted to proven and probable cases (excluding possible cases) presents similar results.

Query
Why do we need paragraph 2.5? It explains the diagnostic criteria (EORTC/MEG, CAPA, AspICU, Bulpa), which are perfectly depicted in the respective publications cited by the authors. Some information are even misleading or at least irritating for the reader: what does the asterisk indicate (L.120)? Why are case definitions explained, which are stated to be not relevant for this study (“not available in this study” / “not performed in this study”) (Ll.115-116, Ll.118-119).

Answer
The criteria and definitions of Aspergillosis are important in this manuscript and although the corresponding articles are cited we believe that briefly describing the specific criteria of each classification may help the understanding of the work. Given that PCR is a new criterion in the EORTC/MSG guidelines, but was not in the previous edition, we consider it relevant to mention that it was not performed in our study. The same for miniBAL

Query
The impact of the modification of the CAPA criteria remains unclear. How many cases were classified as CAPA, which otherwise would be not, based on this modification? What would be the result of this manuscript’s key messages when those cases would have to be excluded from the PPP group? Would we still get the overall same results?
Answer
This study does not propose to modify the criteria for possible IPA, but the possible CAPA classification includes a test (non-bronchoscopic lavage) that very few centers perform and is not sufficiently validated. The manuscript confirms similar results and prognosis, at least in patients with SARS-COV-2 infection, when analyzing PPP Aspergillosis and PP Aspergillosis, highlighting that isolation of Aspergillus in "respiratory samples" should promote early initiation of treatment at least in patients with SARS-COV-2 infection.

Minor comments
Query
L.35: “confirmed independent variables” please rephrase
Answer
The sentence has been changed: Multivariate analysis confirmed the following variables to be independently associated with increased mortality

Query
L.39: importance
Answer
The paragraph has been changed: This series shows that the isolation of Aspergillus spp in respiratory samples, whether associated with disease criteria or not, is linked associated with high mortality, especially in patients with SARS-CoV-2 infection and suggests an early initiation of treatment given its high mortality rate

Query
Ll.48-54: The presented data should be summarized. 202?
Answer
We do not know what the reviewer is referring to in this query.

Query
L.155: PPP-IPA: proven, probable, possible – how about putative?
Answer
proven, probable/putative, possible

Query
L.186; how about acute renal insufficiency? Last line of table?
Answer
acute renal insufficiency is integrated in the variable renal insufficiency (acute or chronic

Query
L.201: “severity of colonized patients” please rephrase
Answer
It has been modified by: confirming the poor prognosis of colonized patients

Query
L.214-216: please clarify the sentence.
Answer
It has been modified by: Mortality was higher in patients with IPA and platelet counts <100,000/µl (14/18, 77.8%) than patients with IPA and >100,000/µl (38/120, 31.7%), p<0.001.

Query
L.218-219: mortality [was] observed?
Answer
It has been modified by: were confirmed

Query
Ll271-272: please rephrase
Answer
Table 5 shows the characteristics of the 31 patients SARS-COV-2 infection with available necropsy studies

Query
L.279: proven instead of proved
Answer
Proven

Query
Ll.302-303: please add a verb.
Answer
Following the SARS-CoV-2 pandemic several published series have highlighted the importance of Aspergillus superinfection in these patients

Query
Ll.303-304: how about reference 29?
Answer
However, few series have analyzed...

Query
Ll.313-315: “the prognosis is worse in infected patients” vs. “the similar results (even higher mortality in colonized patients” seems to be contradictive.
Answer
the 100% mortality rate in colonized patients has been recalculated to 66%. The paragraph has been rewritten: The high mortality in colonized patients in this series are mainly due to patients with SARS-CoV-2 infection. The 66% mortality rate in colonized and SARS-CoV2-infected patients is striking, in contrast to the low mortality rate (~13%) of those colonized in patients without SARS-COV-2 infection.

Query
Ll.414-431: another limitation: single center design
Answer
The sentence has been completed: This study has several limitations. The first is the retrospective nature of the study and the single center design

Reviewer 4 Report

In this retrospective cohort study, Fortun et al. have analyzed 240 hospitalized patients with or without COVID-19 and with positive Aspergillus cultures from any respiratory sample during the defined study period.

They proceed to classify patients as having IPA or Aspergillus colonization, according to whether they fulfill criteria for IPA or not. The classifications used are modified ECMM/ISHAM 2020 consensus criteria (with sputum and tracheobronchial aspirate cultures substituting) for patient with COVID-19, EORTC/MSG criteria for immunocompromised patients, the AspICU criteria for those admitted to ICU and the Bulpa criteria for patients with COPD. According to the criteria used, patients were further classified as proven. probable/putative and possible cases of IPA. Patients not fulfilling these criteria were assigned to the colonization group.

The authors found that 140/240 included patients fulfilled proven/probable/putative/possible IPA criteria, as opposed to 100/240 patients with colonization. Subsequently, the authors describe the differences in baseline characteristics, clinical characteristics, treatment and outcomes for these groups, with no significant differences in mortality, but longer length of stay in hospital and ICU. Logistic regression analysis demonstrates increasing age, renal insufficiency, thrombocytopenia, inotropic support and COVID-19 as independent predictors of mortality, but not the presence of IPA.

Next, the authors describe compare patients with IPA with and without COVID-19 and find no significant differences in mortality, but do in many clinical characteristics and length of stay. Logistic regression analysis reveals the same independent predictors for mortality as in the total population, except increasing age and COVID-19.

In the next section, the authors describe the mycological and radiological findings (as well as IPA classification) of 31 patients who underwent autopsy, four of which had proven IPA on autopsy. Interestingly, 3/27 patients without evidence of IPA at autopsy had demonstrated evidence of IPA before death.

Last, Fortun et al. describe the diagnostic performance of the mycological techniques used in this study for both CAPA and IPA not associated with COVID.

I have read the manuscript with great interest and believe this topic is of major importance, especially concerning the clinical implication/impact of Aspergillus colonization and a "possible" diagnosis of CAPA. The authors have performed a large amount of work retrospectively assessing the patients with respiratory samples positive for Aspergillus and provide a large body of data in their manuscript.

However, I do have several major and some minor remarks about the manuscript. The major comments I have are predominantly related to the study design; to address these, I believe major changes would need to be made to design and analysis:

Major:

1. The authors use several different classification criteria for the diagnosis of IPA for different patient populations in their study (ECMM/ISHAM for COVID, EORTC/MSG for immunocompromised etc.); subsequently, these patients are all pooled into one group of IPA or colonization; although these classifications have been developed for very specific populations indeed, I do not feel they can be pooled in this manner, since the populations are so inherently different and these different classification shave not been compared to one another directly; furthermore, which classification system takes precedence if a patient falls within more than pone category (e.g., has COPD and is immunocompromised): this is not clearly defined by the authors. Could the authors please comment on this?

2. The different classifications have not always been strictly applied according to their original criteria; this is clearly explained for the CAPA criteria (including sputum and TBA cultures instead of NBL cultures), but there are several other adaptations made (I am unsure whether this was done consciously or not, but if so, this should be clearly mentioned): 

- ECMM/ISHAM: originally developed for ICU populations, not for all hospitalized patients; BAL GM => 1.0 defined as positive, authors state GM > 1.0 positive; for tracheobronchitis: was a distinction between probable and possible made here as well? Was GM used as a mycological criterion here as well?

- AspICU: GM is not utilized in the original AspICU criteria; furthermore, original criteria state requirement of either BAL positive for Aspergillus or a classical host risk factor (EORTC) besides a positive respiratory sample for Aspergillus

- Bulpa: How did the authors define a positive serum GM here? I agree that the original paper does not provide a cut-off value, but what did the authors choose here?

Could the authors please comment on these discrepancies? Perhaps some conscious choices were made to amend the criteria, but please clearly state that in the Methods section.

3. A major issue in many papers in my opinion is that patients who have not been tested for IPA according to the applied criteria, are frequently classified as "no IPA"; however, it would be better to say these patients cannot be properly classified: have all the patients in the study undergone appropriate testing to be able to classify them? For possible CAPA this would be no issue, since all included patients will have a positive respiratory sample for Aspergillus; however, for probable CAPA and other classifications: how many patients underwent BAL, BAL culture, BAL GM and serum GM testing? 

4. Directly relating to point 3., I believe a descriptive table would be helpful, indicating how many patients had positive sputum culture, TBA culture, BAL culture, BAL GM etcetera. This might also be helpful in getting insight into why the mortality in COVID+ patients with colonization is 100% and higher that in COVID+ patients with IPA (which is an odd finding)

5. Inclusion of patients who underwent autopsy: for me it is somewhat unclear on what basis these patients were included in the study: all COVID-19 patients who underwent autopsy during the study period? Were patients who had positive Aspergillus cultures in respiratory samples and subsequently underwent autopsy included in both study populations? If so, how many? In other words: was this truly a separate population or part of the greater study population? It now feels somewhat "out of the blue" to mention this in the inclusion criteria

6. Table 1 (line 186) and table 3 (line 246): how can patients fulfill criteria for IPA if they have no infiltrate ion imaging (except when proven diagnosis is found, all classifications require abnormal imaging)

7. The finding COVID+ patients that colonization is associated with 100% mortality and IPA with "only" 40.7% is surprising and counter-intuitive: could the authors comment more on this finding? Could the ECMM/ISHAM classification be flawed?

8. It is currently unclear to me on what basis the authors selected their variables for multivariate logistic regression analysis: were these pre-selected; if so, on what basis? Or were all variables also included in multivariate analysis? In that case, there is a risk of overfitting, since the number of selected variables seems high compared to the n of the outcome of interest (here: mortality).

9. Paragraph 3.5 and table 6 (lines 281 and 298, respectively): Could the authors please clarify what the gold standard  for diagnosis is here? If this is the ECMM/ISHAM classification, the authors would compare the diagnostic performance of the tests against a classification partially based on that test, which would not be logical. An independent method of ascertaining the diagnosis would be needed; also, the n for some tests is very low. Since testing diagnostic performance seems somewhat out of place in this study,  is not really mentioned in the study objectives and, in my personal opinion, distracts somewhat from the overall message, I would consider leaving out this section and this table entirely.

Minor:

1. Exclusion criteria (line 92 - 106): mentions absence of inclusion criteria and is somewhat superfluous; furthermore, absence of SARS-CoV-2 infection is an exclusion criterion mentioned here: this does not seem to be correct, as COVID negative patients were included as well. Please explain or adjust.

2. Table 1 (line 186): Please define "oxygen high requirements"; also, alignment is off

3. Line 197: does the similar mortality mentioned here pertain to PPP-IPA itself or colonization according to PPP-IPA criteria? In the latter case, I agree with the statement, but in the former, a difference between 25% and 14% I would not call very similar

4. Table 2 and 4: It is unclear to me what the values in the second column represent exactly: mortality rates in case patient fulfills the binary value in the first column? And what do the numerical values mean here? Why not show the OR values in univariate analysis in these tables?

5. Line 222 - 224: Please rephrase, looks confusing now (as if 57.1% is lower than 17.4%)

6. Sometimes difficult to follow sentences (e.g., see line 189-190), please consider a native speaker reviewing the paper

7. Please check for typos or untranslated text (e.g., "tracheobronquitis") and minor spelling or interpunction mistakes

Author Response

Thank you very much for your attention and interest in our manuscript.
This new version is uploaded to the digital platform. The following modifications are detailed point by point and highlighted in yellow in the text.

Reviewer: 4

In this retrospective cohort study, Fortun et al. have analyzed 240 hospitalized patients with or without COVID-19 and with positive Aspergillus cultures from any respiratory sample during the defined study period.
They proceed to classify patients as having IPA or Aspergillus colonization, according to whether they fulfill criteria for IPA or not. The classifications used are modified ECMM/ISHAM 2020 consensus criteria (with sputum and tracheobronchial aspirate cultures substituting) for patient with COVID-19, EORTC/MSG criteria for immunocompromised patients, the AspICU criteria for those admitted to ICU and the Bulpa criteria for patients with COPD. According to the criteria used, patients were further classified as proven. probable/putative and possible cases of IPA. Patients not fulfilling these criteria were assigned to the colonization group.
The authors found that 140/240 included patients fulfilled proven/probable/putative/possible IPA criteria, as opposed to 100/240 patients with colonization. Subsequently, the authors describe the differences in baseline characteristics, clinical characteristics, treatment and outcomes for these groups, with no significant differences in mortality, but longer length of stay in hospital and ICU. Logistic regression analysis demonstrates increasing age, renal insufficiency, thrombocytopenia, inotropic support and COVID-19 as independent predictors of mortality, but not the presence of IPA.
Next, the authors describe compare patients with IPA with and without COVID-19 and find no significant differences in mortality, but do in many clinical characteristics and length of stay. Logistic regression analysis reveals the same independent predictors for mortality as in the total population, except increasing age and COVID-19.
In the next section, the authors describe the mycological and radiological findings (as well as IPA classification) of 31 patients who underwent autopsy, four of which had proven IPA on autopsy. Interestingly, 3/27 patients without evidence of IPA at autopsy had demonstrated evidence of IPA before death.
Last, Fortun et al. describe the diagnostic performance of the mycological techniques used in this study for both CAPA and IPA not associated with COVID.
I have read the manuscript with great interest and believe this topic is of major importance, especially concerning the clinical implication/impact of Aspergillus colonization and a "possible" diagnosis of CAPA. The authors have performed a large amount of work retrospectively assessing the patients with respiratory samples positive for Aspergillus and provide a large body of data in their manuscript.
However, I do have several major and some minor remarks about the manuscript. The major comments I have are predominantly related to the study design; to address these, I believe major changes would need to be made to design and analysis:

Major:
Query
1. The authors use several different classification criteria for the diagnosis of IPA for different patient populations in their study (ECMM/ISHAM for COVID, EORTC/MSG for immunocompromised etc.); subsequently, these patients are all pooled into one group of IPA or colonization; although these classifications have been developed for very specific populations indeed, I do not feel they can be pooled in this manner, since the populations are so inherently different and these different classification shave not been compared to one another directly; furthermore, which classification system takes precedence if a patient falls within more than pone category (e.g., has COPD and is immunocompromised): this is not clearly defined by the authors. Could the authors please comment on this?
Answer
This is a very relevant assessment. Indeed, as several classifications were used, patients were assigned to each group (private, probable/putative, possible) according to their specific condition and classification. In case of belonging to more than one group they were prioritized as follows: In patients with SARS-COV-2 infection, the modified ECMM/ISHAM 2020 consensus was always applied. In patients without SARS-COV-2 infection, the EORTC/MSG criteria were taken into account first, then the AspICU criteria and finally the Bulpa criteria. This clarification has been included in the text

Query
2. The different classifications have not always been strictly applied according to their original criteria; this is clearly explained for the CAPA criteria (including sputum and TBA cultures instead of NBL cultures), but there are several other adaptations made (I am unsure whether this was done consciously or not, but if so, this should be clearly mentioned):
Answer
Sputum and TBA cultures instead of NBL cultures were only considered in patients with SARS-COV-2 infection according to the modified ECMM/ISHAM 2020 consensus. This clarification has been included in the text

Query
- ECMM/ISHAM: originally developed for ICU populations, not for all hospitalized patients; BAL GM => 1.0 defined as positive, authors state GM > 1.0 positive; for tracheobronchitis: was a distinction between probable and possible made here as well? Was GM used as a mycological criterion here as well?
Answer
For homogenization purposes, GM was considered positive in serum if >0.5 and in BAL if >1.0.

Query
- AspICU: GM is not utilized in the original AspICU criteria; furthermore, original criteria state requirement of either BAL positive for Aspergillus or a classical host risk factor (EORTC) besides a positive respiratory sample for Aspergillus
- Bulpa: How did the authors define a positive serum GM here? I agree that the original paper does not provide a cut-off value, but what did the authors choose here?
Answer
For homogenization purposes, GM was considered positive in serum if >0.5 and in BAL if >1.0.

Query
3. A major issue in many papers in my opinion is that patients who have not been tested for IPA according to the applied criteria, are frequently classified as "no IPA"; however, it would be better to say these patients cannot be properly classified: have all the patients in the study undergone appropriate testing to be able to classify them? For possible CAPA this would be no issue, since all included patients will have a positive respiratory sample for Aspergillus; however, for probable CAPA and other classifications: how many patients underwent BAL, BAL culture, BAL GM and serum GM testing?
Answer
We agree with the editor's annotation. Since this was a retrospective study, the availability of radiological and microbiological diagnostic techniques was not 100% available in all patients and could have led to a classification bias. The manuscript discusses these limitations, although the large number of patients included and the confirmation that the results in PPP-IPA are very similar to those obtained in PP-IPA corroborate the results.

Query
4. Directly relating to point 3., I believe a descriptive table would be helpful, indicating how many patients had positive sputum culture, TBA culture, BAL culture, BAL GM etcetera. This might also be helpful in getting insight into why the mortality in COVID+ patients with colonization is 100% and higher that in COVID+ patients with IPA (which is an odd finding)
Answer
the 100% mortality rate in colonized patients has been recalculated to 66%. The paragraph has been rewritten: The high mortality in colonized patients in this series are mainly due to patients with SARS-CoV-2 infection. The 66% mortality rate in colonized and SARS-CoV2-infected patients is striking, in contrast to the low mortality rate (~13%) of those colonized in patients without SARS-COV-2 infection. The table suggested by the reviewer could be interesting, but we believe that the work is already sufficiently extensive in text and tables (n: 6) to be able to include another one. Much of the information suggested by the reviewer is specified in the text.

Query
5. Inclusion of patients who underwent autopsy: for me it is somewhat unclear on what basis these patients were included in the study: all COVID-19 patients who underwent autopsy during the study period? Were patients who had positive Aspergillus cultures in respiratory samples and subsequently underwent autopsy included in both study populations? If so, how many? In other words: was this truly a separate population or part of the greater study population? It now feels somewhat "out of the blue" to mention this in the inclusion criteria
Answer
We thought it relevant to contribute our experience given the lack of data with necropsy studies in patients with SARS-COV-2 infection. Of the 31 patients, only four showed confirmatory findings of IPA and they are included in the study as proven IPA. In the 27 patients in whom autopsy did not show findings suggestive of IPA, three had shown before death a diagnosis of IPA (2 of them PP-IPA) and they were included in the study. The remaining 24 patients are not included in the IPA study.

Query
6. Table 1 (line 186) and table 3 (line 246): how can patients fulfill criteria for IPA if they have no infiltrate ion imaging (except when proven diagnosis is found, all classifications require abnormal imaging)
Answer
The 2 patients without infiltrates included in the IPA group correspond to patients with tracheobronchitis. Although it may be questionable to include them, we decided to do so because they were patients suffering from a certain form of aspergillosis.

Query
7. The finding COVID+ patients that colonization is associated with 100% mortality and IPA with "only" 40.7% is surprising and counter-intuitive: could the authors comment more on this finding? Could the ECMM/ISHAM classification be flawed?
Answer
The 100% mortality rate in colonized patients has been recalculated to 66%. The paragraph has been rewritten: The high mortality in colonized patients in this series are mainly due to patients with SARS-CoV-2 infection. The 66% mortality rate in colonized and SARS-CoV2-infected patients is striking, in contrast to the low mortality rate (~13%) of those colonized in patients without SARS-COV-2 infection.

Query
8. It is currently unclear to me on what basis the authors selected their variables for multivariate logistic regression analysis: were these pre-selected; if so, on what basis? Or were all variables also included in multivariate analysis? In that case, there is a risk of overfitting, since the number of selected variables seems high compared to the n of the outcome of interest (here: mortality).
Answer
For the multivariate studies, all variables that had shown p<0.05 in the univariate study were included. We also included variables that were associated with significance in the literature (e.g., age, SARS-COV-2 infection, etc.).
Query
9. Paragraph 3.5 and table 6 (lines 281 and 298, respectively): Could the authors please clarify what the gold standard  for diagnosis is here? If this is the ECMM/ISHAM classification, the authors would compare the diagnostic performance of the tests against a classification partially based on that test, which would not be logical. An independent method of ascertaining the diagnosis would be needed; also, the n for some tests is very low. Since testing diagnostic performance seems somewhat out of place in this study,  is not really mentioned in the study objectives and, in my personal opinion, distracts somewhat from the overall message, I would consider leaving out this section and this table entirely.
Answer
We can understand the reviewer's concern given the heterogeneity of the patients. To control for this heterogeneity the table gives 4 types of analysis (with and without SARS-COV-2 infection and PPP-IPA vs PP-IPA). As mentioned above, in patients with SARS-COV-2 infection, the modified ECMM/ISHAM 2020 consensus was applied in priority in all of them, which gives homogeneity to the sample. On the other hand, the inclusion of PP-IPA (excluding possible IPA) also homogenized the results in patients with and without SARS-COV-2 infection.

Minor:
Query
1. Exclusion criteria (line 92 - 106): mentions absence of inclusion criteria and is somewhat superfluous; furthermore, absence of SARS-CoV-2 infection is an exclusion criterion mentioned here: this does not seem to be correct, as COVID negative patients were included as well. Please explain or adjust.
Answer
This is a mistake. Absence of SARS-CoV-2 infection is not an exclusion criterion. This sentence has been deleted

Query
2. Table 1 (line 186): Please define "oxygen high requirements"; also, alignment is off
Answer
high-flow oxygen or use of a reservoir.

Query
3. Line 197: does the similar mortality mentioned here pertain to PPP-IPA itself or colonization according to PPP-IPA criteria? In the latter case, I agree with the statement, but in the former, a difference between 25% and 14% I would not call very similar
Answer
The paragraph has been modified: However, in patients without SARS-COV-2 infection, mortality in colonized patients was 13.2% (10/76) and 25% (8/32) in the IPA group (p = 0.16). When possible IPA was excluded and only proved/probable or putative IPA was considered (PP-IPA), among colonized patients the mortality rate remained high (50%) in patients with SARS-COV-2 infection and was lower in patients without SARS-COV-2 infection (14.0%), similar to mentioned 13.2%.

Query
4. Table 2 and 4: It is unclear to me what the values in the second column represent exactly: mortality rates in case patient fulfills the binary value in the first column? And what do the numerical values mean here? Why not show the OR values in univariate analysis in these tables?
Answer
Due to space problem OR values in univariate analysis in these tables are not shown. P shows the comparison of the variables in relation to overall mortality. In the multivariate analysis (more relevant) p and OR are shown.

Query
5. Line 222 - 224: Please rephrase, looks confusing now (as if 57.1% is lower than 17.4%)
Answer
It has been modified by: Mortality was higher in patients with IPA and platelet counts <100,000/µl (14/18, 77.8%) than patients with IPA and >100,000/µl (38/120, 31.7%), p<0.001.

Query
6. Sometimes difficult to follow sentences (e.g., see line 189-190), please consider a native speaker reviewing the paper
Answer
It has been modified by There were more colonized patients in SARS-CoV-2 group (76 colonized and 32 IPA) than in patients without SARS-CoV-2 infection (24 colonized, 108 IPA).

Query
7. Please check for typos or untranslated text (e.g., "tracheobronquitis") and minor spelling or interpunction mistakes
Answer
The entire manuscript has been revised and proofread by professional Native American translators (American Manuscripts Editors).

Round 2

Reviewer 1 Report

Thank you for the revised version. Although improved, I feel my original concerns have not been addressed.

Round 2

Thank you again for your interest in the article.

I am sending you a new version of the manuscript incorporating in yellow all the changes and trying to respond to the reviewers' queries. I am sending you the reply to the reviewers' queries.

Query:

Reviewer 1.

Thank you for the revised version. Although improved, I feel my original concerns have not been addressed.

     Answer:

We have reviewed and corrected the reviewer 1's queries. We recommend reading the commentary to the last query of reviewer 3 because they agree on the doubts about the methodology that explains the differences between colonization and possible aspergillosis in patients with SARS-CoV-2 infection.

Reviewer 3 Report

Thank you for the replies.

Major comments

It was the very major criticism of the original manuscript that it was sloppily written. It is very bad style when scientific reviewers are misused as proofreaders. It is even worse when their findings are ignored. It is worst when the authors still claim to have corrected the issues.

Of the ten queries of the first review, the following are still to be addressed (numeration of last review applied):

2. It is not COVID but COVID-19.

3. The authors reacted as follows: “The following has been standardized and is used throughout the manuscript: SARS-COV-2 or SARS-COV-2 infection.”

1.       No. It is not used throughout the manuscript. Not even close. There are at least three different variants included: “SARS-COV-2”, “SARS-CoV-2”, and “SARS-Cov2”.

2.       There is a common nomenclature for this virus. It is not “SARS-COV-2”.

4. Still Aspergillus is not in italic.

5. IAPA is Influenza associated pulmonary aspergillosis but not Influenza associated aspergillosis.

8. In the material and methods section, the applied tests kits are described. While the PCR is specified by stating name, manufacturer, and country (“QuantStudio S5 Real-time PCR system; ThermoFisher, USA”), the antigen test misses one of these information (“Platelia™ Aspergillus; Bio-Rad Laboratories”). This should be consistent. It still isn’t.

10. “Several times, there is a sudden change to other fonts or font sizes.” For instance, please check the references (underlinings…).

Another major criticism of the original manuscript (citated from the last review): “Several comparisons between the IPA and the colonization cohort should be considered with caution, since there is a severe risk of bias. For instance, the IPA cohort was described to feature radiological findings more frequently compared to the colonization cohort. This is misleading since the IPA cohort is (at least for the majority [?] of its cases) defined by the presence of those radiological findings. In other words: many of the respective cases of the colonization cohort are only included in this cohort because they do not have those radiological findings and would become part of the IPA cohort as soon as they had those radiological findings (EORTC/MSG criteria).”

The problem is the comparison between probable IPA and colonization. If there are two patients with the same risk factor and with the same microbiological finding (growth of Aspergillus in the airways), then there is, according to the EORTC/MEG criteria, only one factor left that is decisive for the categorization as either probable IA or colonization: the radiological finding. Colonized patients with risk factors are only one CT from being probable IA. As soon as the CT is positive for IPA, the colonization patient becomes a probable IPA patient. Therefore, a comparison of radiology between the two subgroups is very risky in terms of bias (independent variable???).

However, the answer of the authors is absolutely not satisfying (and also not very clear…): “Yes. This aspect has also been discussed with previous reviewers. Limitations between possible and colonized cases, especially in SARS-COV-2 infected patients.” Possible IPA is not even involved in my criticism, since possible IA cannot even be colonization - growth of Aspergillus is “mycological evidence” according to EORTC/MSG criteria, which promotes a patient from possible to probable IA.

The respective issue has not been addressed.

Author Response

Reviewer 3.

Major comments

It was the very major criticism of the original manuscript that it was sloppily written. It is very bad style when scientific reviewers are misused as proofreaders. It is even worse when their findings are ignored. It is worst when the authors still claim to have corrected the issues.

Of the ten queries of the first review, the following are still to be addressed (numeration of last review applied):

Query:

  1. It is not COVID but COVID-19.

     Answer:

COVID-19 (in capital letters) has been used throughout the document.

Query:

  1. The authors reacted as follows: “The following has been standardized and is used throughout the manuscript: SARS-COV-2 or SARS-COV-2 infection.”
  2. No. It is not used throughout the manuscript. Not even close. There are at least three different variants included: “SARS-COV-2”, “SARS-CoV-2”, and “SARS-Cov2”.
  3. There is a common nomenclature for this virus. It is not “SARS-COV-2”.

     Answer:

      We apologize for any errors in nomenclature. Following the recommendations in scientific texts, the following nomenclatures have been used throughout the manuscript and tables: SARS-CoV-2.

      Query:

  1. Still Aspergillus is not in italic.

Answer:

The text has been revised and Aspergillushas been edited in italics.

      Query:

  1. IAPA is Influenza associated pulmonary aspergillosis but not Influenza associated aspergillosis.

Answer:

Yes, IAPA is Influenza associated pulmonary aspergillosis and the text has been corrected

      Query:

  1. In the material and methods section, the applied tests kits are described. While the PCR is specified by stating name, manufacturer, and country (“QuantStudio S5 Real-time PCR system; ThermoFisher, USA”), the antigen test misses one of these information (“Platelia™ Aspergillus; Bio-Rad Laboratories”). This should be consistent. It still isn’t.

Answer:

The final text is as follows "Detection of SARS-CoV-2 virus was performed by real-time RT-PCR following the WHO and/or CDC protocol (QuantStudio S5 Real-time PCR system; ThermoFisher, USA). The galactomannan (GM) antigen index was measured with a sandwich enzyme-linked immunosorbent assay (ELISA) (Platelia™ Aspergillus; Bio-Rad Laboratories, Madrid, Spain)"

Query:

  1. “Several times, there is a sudden change to other fonts or font sizes.” For instance, please check the references (underlinings…).

Answer

The manuscript has been entirely revised, including references. Times New Roman 12 is used systematically. The editors have used Palatino Linotype for the title, authors and affiliation on the first page and it have been respected

Query:

Another major criticism of the original manuscript (citated from the last review): “Several comparisons between the IPA and the colonization cohort should be considered with caution, since there is a severe risk of bias. For instance, the IPA cohort was described to feature radiological findings more frequently compared to the colonization cohort. This is misleading since the IPA cohort is (at least for the majority [?] of its cases) defined by the presence of those radiological findings. In other words: many of the respective cases of the colonization cohort are only included in this cohort because they do not have those radiological findings and would become part of the IPA cohort as soon as they had those radiological findings (EORTC/MSG criteria).”

The problem is the comparison between probable IPA and colonization. If there are two patients with the same risk factor and with the same microbiological finding (growth of Aspergillus in the airways), then there is, according to the EORTC/MEG criteria, only one factor left that is decisive for the categorization as either probable IA or colonization: the radiological finding. Colonized patients with risk factors are only one CT from being probable IA. As soon as the CT is positive for IPA, the colonization patient becomes a probable IPA patient. Therefore, a comparison of radiology between the two subgroups is very risky in terms of bias (independent variable???).

However, the answer of the authors is absolutely not satisfying (and also not very clear…): “Yes. This aspect has also been discussed with previous reviewers. Limitations between possible and colonized cases, especially in SARS-COV-2 infected patients.” Possible IPA is not even involved in my criticism, since possible IA cannot even be colonization - growth of Aspergillus is “mycological evidence” according to EORTC/MSG criteria, which promotes a patient from possible to probable IA.

The respective issue has not been addressed.

Answer

We understand the reviewer's doubts and the weak difference in this work between possible aspergillosis and colonization in patients with SARS-Cov-2 infection. This is due to the small difference as a diagnostic tool between "non-bronchoscopic lavage" recommended by The ECMM/ISHAM 2020 consensus and "non-bronchoscopic sample, including TBA or sputum" used in this work. On the other hand, this aspect is an important point of discussion of the article, because in patients with SARS-Cov-2 infection mortality in colonized patients is very high: whether they are considered as colonized or as possible, and similar to that observed in probable cases (where only diagnoses obtained by bronchoscopy are taken into account). However, to better define the concept of possible aspergillosis in patients with SARS-CoV-2 infection and to differentiate them from colonized patients, we have included the following concept in methods:

Possible CAPA:Compatible clinical and imaging findings, with microbiological isolation of Aspergillusin respiratory specimens other than BAL:   TBA, sputum. Possible pulmonary CAPA requires pulmonary infiltrates,         well-circumscribed lesions(s) or nodules, preferably documented by chest CT, or             cavitating infiltrate, which is not attributed to another cause. In patients with     bilateral, ground-glass opacities or other COVID-19 related findings, significant radiological changes as previously mentioned and confirmed by an expert radiologist were requiredto be considered possible CAPA. The ECMM/ISHAM 2020 consensus includes non-bronchoscopic lavage as a diagnostic tool (*).Non-bronchoscopic lavage was not performed in this study In this series, these were replaced (only in patients with SARS-CoV-2 infection) with non-bronchoscopic samples: TBA, sputum.

Colonization was considered in patients with no radiological findings o unchanged with respect to those attributed to COVID-19.
